# ROUTERARENA: AN OPEN PLATFORM FOR COMPREHENSIVE COMPARISON OF LLM ROUTERS

**Yifan Lu**[*], **Rixin Liu**[*], **Jiayi Yuan**[*], **Xingqi Cui, Shenrun Zhang, Hongyi Liu, Jiarong Xing**
Department of Computer Science, Rice University, Houston, TX 77005, USA
{yl231, rl165, jy101, xc66, sz81, hl87, jx22}@rice.edu

## ABSTRACT

Today's LLM ecosystem comprises a wide spectrum of models that differ in size, capability, and cost. No single model is optimal for all scenarios; hence, LLM routers have become essential for selecting the most appropriate model under varying circumstances. However, the rapid emergence of various routers has led to fragmented evaluation practices and inconsistent metrics, making it difficult to systematically assess progress in this space. To address this problem, we need a comprehensive router comparison and a standardized leaderboard, similar to those available for models. In this work, we introduce ROUTERARENA, the first open platform enabling *comprehensive* comparison of LLM routers. ROUTERARENA has (1) a principally constructed dataset with broad knowledge domain coverage, (2) distinguishable difficulty levels for each domain, (3) an extensive list of evaluation metrics, and (4) an automated framework for evaluation and leaderboard updates. Leveraging this framework, we have produced the initial leaderboard with detailed metrics comparison. Figure 1 provides a preview of the leaderboard. The complete framework and the latest router leaderboard are publicly available at https://routeworks.github.io/.

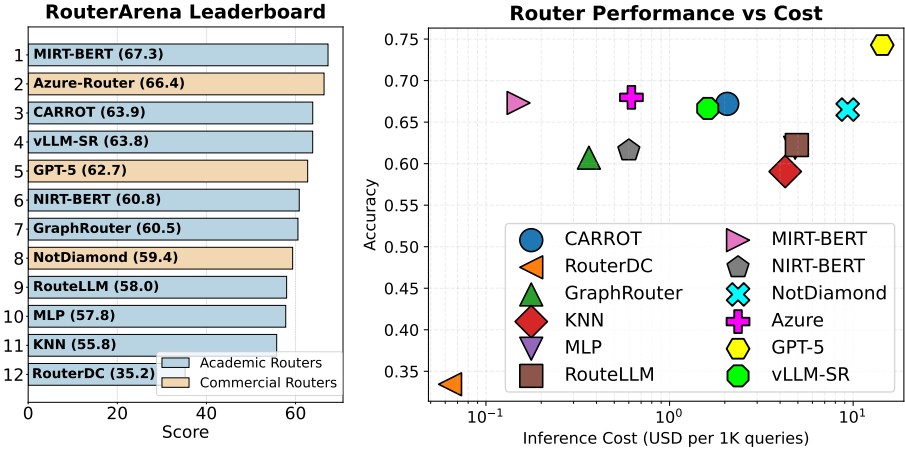

Figure 1: A preview of ROUTERARENA's leaderboard and performance-cost trade-off comparison.

## 1 INTRODUCTION

Large Language Models (LLMs) are rapidly diversifying, offering an ever-wider spectrum of capabilities at different inference costs. This diversity increasingly challenges the prevailing LLM usage pattern in which users manually choose models for their queries. The difficulty stems from the fact that no single model is universally optimal: powerful models excel at complex tasks but are costly, while smaller models are more efficient yet may struggle on difficult queries. As a result, *LLM routers* that automatically select models based on input queries are increasingly recognized as a core system primitive in practical deployments.

---

[*]These authors contributed equally to this work.

Given its importance and promise, many LLM routers have recently emerged in both industry and academia (Figure 2). A notable example is GPT-5 (OpenAI, 2025), which incorporates routing as a key feature by directing user queries to the most suitable model within the OpenAI family. As routing rapidly proliferates and becomes a core function in LLM systems, the evaluation frontier correspondingly shifts: beyond comparing individual models, we must now rigorously evaluate routing strategies. Unfortunately, **router evaluation has not kept pace: there is currently no open evaluation platform**, akin to LMArena (Chiang et al., 2024), that systematically compares open-source routers (Hu et al., 2024; Zhuang et al., 2024) and commercial routing services (NotDiamond, 2025; Microsoft, 2025) under a unified protocol.

It is urgent to fill this gap by building a *Router Arena* that can comprehensively evaluate and compare routers, providing a clear and systematic view of the state of the art. However, unlike model arenas, designing a router arena is considerably more challenging due to the requirements from three key aspects. (1) **Dataset**. To evaluate whether a router can recognize problem domains and dispatch queries to appropriate models at minimal cost, the arena dataset must cover a broad range of domains and subjects, as well as varying difficulty levels. (2) **Metrics**. Router performance is inherently multi-dimensional, and so should be the arena ranking. While accuracy and cost are the primary metrics, it is also important to capture other dimensions such as routing optimality and robustness. (3) **Framework**. To enable live leaderboard updates, the arena must have a user-friendly framework that can automatically evaluate new open-source and commercial routers. Although existing studies explore some of these directions, as summarized in Table 1 and discussed in Section 2, they fail to address each challenge in a *comprehensive* way.

In this work, we present ROUTERARENA, the first open platform for comprehensive evaluation and comparison of LLM routers. It addresses the above key challenges with the following designs:

- **A Principled Diverse Dataset.** To ensure broad coverage, we construct the dataset using the Dewey Decimal Classification system adopted in libraries, covering all domains except religion. For each subject, we apply Bloom's taxonomy to ensure queries span diverse cognitive skills and empirically validate that the queries cover three difficulty levels. This produces a diverse dataset of ~8,400 queries spanning 9 domains and 44 categories for router evaluation.
- **Extensive Metrics for Arena Ranking.** We construct router leaderboards by considering an extensive list of deployment-relevant metrics including query-answer accuracy, query-answer cost, routing optimality (cheapest correct selection), robustness to query perturbations (consistency), and router overhead (latency). This enables router comparison from multiple perspectives.
- **An Automated Framework for Leaderboard Updates.** We design a framework that automatically evaluates new routers, collects metrics, and updates the leaderboard. The framework supports both open-source and commercial routers, and employs prefix caching to improve efficiency.

Figure 1 provides a preview of our accuracy-cost leaderboard along with other details. We have found that although GPT-5 achieves higher accuracy, its cost is significantly higher than that of other routers due to its model pool being restricted to the OpenAI family. Consequently, it does not rank as the best router on our accuracy-cost leaderboard.

Our vision is for ROUTERARENA to serve as an open community venue for evaluating routers as the ecosystem evolves, providing a standardized basis for fair comparison and progress tracking. By lowering the barrier to evaluation and enabling transparent, reproducible results, ROUTERARENA will help researchers and practitioners design, improve, and adopt better routers.

## 2 MOTIVATION

**The Rapid Emergence of LLM Routers.** As shown in Figure 2, the landscape of LLM routers is rapidly expanding, evolving from academic exploration to commercial deployment. From a few scattered academia routers (Zhang et al., 2023; Chen et al., 2023; Hari & Thomson, 2023) in mid-2023, the number of publications expanded to more than a dozen by 2024 (Liu et al., 2024; Zhuang et al., 2024; Chen et al., 2024; Zhao et al., 2024). By 2025, not only did academia routers continue to grow (Wang et al., 2025; Huang et al., 2025; Ding et al., 2025; Zhang et al., 2025b), but commercial products also emerged (NotDiamond, 2025; Microsoft, 2025), most notably GPT-5 (OpenAI, 2025) with a built-in router.

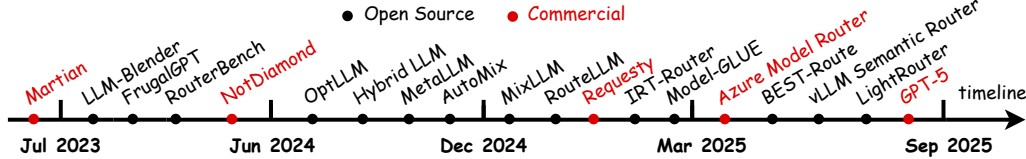

Figure 2: Timeline of example router-related work and products.

Table 1: Comparison of existing work (Hu et al., 2024; Huang et al., 2025; Feng et al., 2025b; Zhuang et al., 2024) and ROUTERARENA. ROUTERARENA enables comprehensive router comparison with extensive query categories, difficulty levels, evaluation metrics, and router inclusion.

| Benchmark | Query categories | Difficulty levels | Evaluation Metrics | Commercial Routers | Router Ranking |
|---|---|---|---|---|---|
| RouterBench | 24 Categories | ✗ No analysis | ✗ Deferral curve only | ✗ | ✗ |
| RouterEval | 27 Categories | ✗ No analysis | ✗ Accuracy metric only | ✗ | ✗ |
| FusionBench | 26 Categories | LLM-judge analysis | ✗ Deferral curve only | ✗ | ✗ |
| EmbedLLM | 26 Categories | ✗ No analysis | ✗ Accuracy metric only | ✗ | ✗ |
| ROUTERARENA | 44 Categories based on DDC | ✓ 3 Empirically-Verified Difficulty Levels | ✓ 5 Evaluation perspectives | ✓ 3 Included | ✓ Multi-metric leaderboard |

**New Problem: The Need for Systematic Router Evaluation.** With the rapid proliferation of routing methods, the field lacks a principled framework to comprehensively evaluate and compare them. Such comprehensiveness entails dataset categories, difficulty levels, evaluation metrics, and router inclusion. However, our review of existing work reveals that **no existing benchmark provides such holistic evaluation**. As shown in Table 1, prior work falls short in the following aspects.

- *Narrow Query Category Coverage.* They lack full coverage of query categories, making them impossible to evaluate router performance on queries from excluded categories.
- *Indistinguishable Difficulty Levels.* They do not differentiate queries by difficulty, limiting their ability to test accuracy-cost tradeoffs.
- *Limited Evaluation Metrics.* They only consider a subset of relevant metrics, overlooking important dimensions such as optimality, robustness, and latency.
- *No Support of Commercial Routers.* Current frameworks evaluate only open-source routers and do not extend to closed-source or commercial routers.
- *No Router Leaderboard.* There is no leaderboard that allows people to compare all routers under a unified evaluation protocol.

**This Work: ROUTERARENA.** This gap motivates us to design ROUTERARENA, an open platform for comprehensive router comparisons. In the remainder of this paper, we first introduce the key components of ROUTERARENA: principled dataset construction, comprehensive metric formulation, and an automated evaluation framework with live leaderboard. We then present our evaluation results and discuss the key findings.

## 3 ROUTERARENA EVALUATION DATASET

To enable meaningful and unbiased router comparisons, a high-quality evaluation dataset is essential. In this work, we construct such a dataset by adhering to three guiding principles.

**Principle 1: DDC-Inspired Diverse Domain Coverage.** To evaluate a router's ability to recognize problem domains and route queries to the appropriate specialist models, the dataset must provide broad domain coverage. To achieve this, we draw inspiration from the Dewey Decimal Classification (DDC) system (Dewey, 1876), a book classification framework widely used in libraries. The DDC is renowned for its comprehensive and logical structure, providing a proven methodology for organizing the entire world of knowledge into distinct, hierarchical categories (Satija, 2013).

**Principle 2: Bloom-Guided Cognitive Skill Levels.** Routers in practice face queries that range from simple factual recall to complex reasoning. To ensure coverage of *cognitive skills* from simple recall to higher-order reasoning and judgment, we adopt Bloom's taxonomy (Bloom et al., 1956), a widely used framework for quantifying question complexity and cognitive processes (Ullrich & Geierhos, 2021; Herrmann-Werner et al., 2024; Padó, 2017).

**Principle 3: Empirically Defined Difficulty Levels.** To test whether a router can make meaningful accuracy-cost tradeoffs—choosing between powerful but expensive models and weaker but cheaper ones—the dataset must also provide clearly distinguishable *difficulty* levels. We define difficulty empirically: for each query, we evaluate it across a diverse set of models and measure the fraction that produce a correct answer. Queries solved by fewer models are considered more difficult.

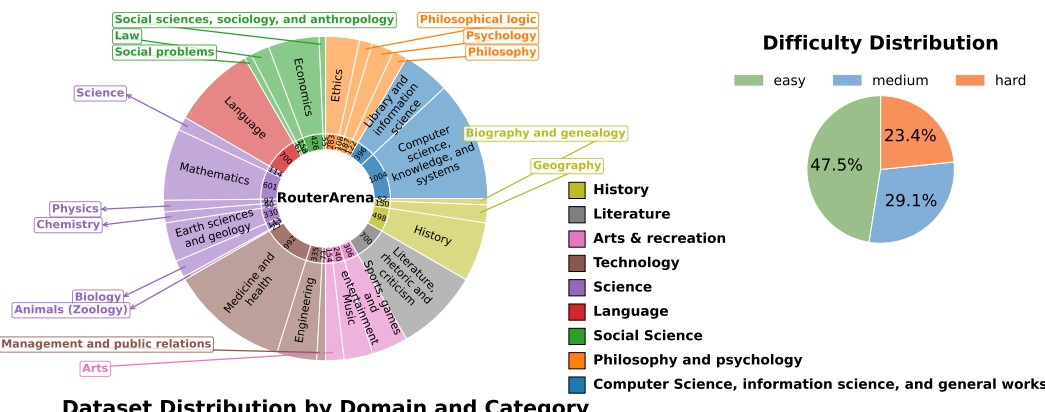

**Dataset Distribution by Domain and Category**

Figure 3: Dataset composition. For ease of demonstration, we merged some categories.

**Dataset Construction Process.** Following these principles, we curate our evaluation dataset as follows. To ensure coverage across all DDC categories (excluding religion), we first collect all the queries from two existing LLM benchmark datasets and then supplement underrepresented categories with data from 23 open-source, domain-specific datasets, resulting in 62,000 raw queries. We then cap each source dataset at approximately 1,000 queries.[1] To determine the cognitive skill level of each query based on Bloom's taxonomy, we employ an LLM-as-Judge approach with DEEPSEEK-V3.1 (DeepSeek-AI, 2025) (prompt specified in Appendix D). We validate the effectiveness of this with a small-scale human study (Appendix C.1). Note that we use these Bloom levels only to characterize the cognitive skills involved in each query; the ground-truth difficulty labels are empirically determined based on Principle 3.

Next, to fairly distribute questions across categories and cognitive levels, we propose a recursive deficit redistribution algorithm. We begin by setting the ratio of science to humanities at 2:1. Within each top-level category, if a sub-category falls short of its proportional quota, the resulting surplus is recursively and uniformly redistributed to those sub-categories that exceed their initial allocation. We apply the same procedure within each sub-category to allocate data across different cognitive levels, ensuring balanced coverage throughout the dataset.

However, this raw dataset contains many highly similar or even duplicate questions inherited from the various sampled sources. Such redundancy does not benefit router evaluation and may even introduce noise into the results. To address this, we perform cosine-similarity-based de-duplication using SENTENCE-TRANSFORMERS/ALL-MINILM-L6-V2. By strictly following the allocation strategy and selecting the least similar samples, we ensure that the resulting ROUTERARENA dataset maintains broad coverage with minimal redundancy.

**The Resulting Dataset.** Our final evaluation dataset consists of 8,400 queries sampled from 23 source datasets. It spans nine top-level domains and 44 categories in Figure 3 (Left). To empirically characterize difficulty, we select 42 representative models covering a broad spectrum of capabilities (details in Appendix C.4). For each query, we measure the number of models that produce a correct answer and categorize queries into three bands—hard ($\leq$4/42), medium (5-19/42), and easy ($\geq$20/42)—as shown in Figure 3 (right). Figure 4 further shows that, when sorted by this measure, both the overall and per-domain curves increase smoothly from low to high accuracy without large gaps, indicating a well-spread spectrum of difficulty that can distinguish router behavior. Note that this empirical difficulty distribution is skewed, but it reflects real-world query patterns—easy and

---

[1]It primarily affects large corpora such as WMT and SuperGLUE and yields roughly 30,000 candidate queries while preventing a few sources from dominating the pool.

medium questions occurring more frequently than hard ones. We include more details about the dataset in Appendix C.2, including dataset schema and concrete query examples.

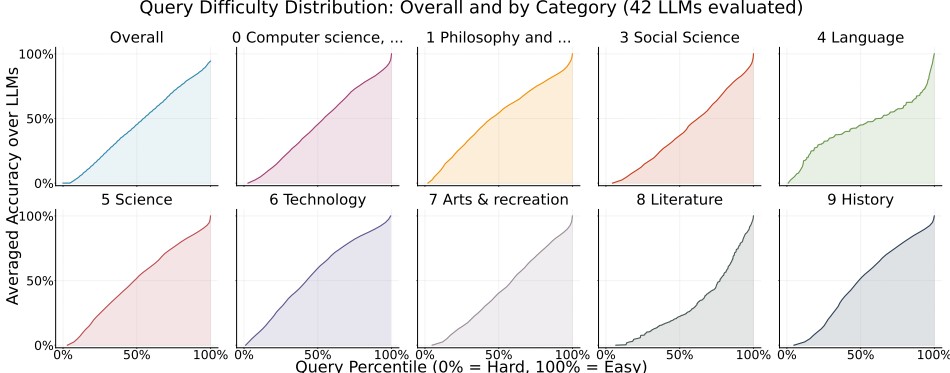

Figure 4: Queries are sorted by their average accuracy over all 42 LLMs. We observe smooth, monotonic curves overall and in each domain, indicating that query difficulty is well spread across the full range from very hard (left) to very easy (right).

# 4 ROUTERARENA EVALUATION METRICS

ROUTERARENA supports comprehensive router evaluation along five dimensions.

**(1) Query-Answer Accuracy.** This metric captures a router's ability to direct queries to the appropriate models such that they are correctly answered. We calculate accuracy as the average correctness across all our dataset queries.

**(2) Query-Answer Cost.** This measures the cost incurred by a router's routing decisions. To address important factors such as the variable cost introduced by input length and generation length (e.g., in chain-of-thought reasoning) as well as the distinct computational characteristics of Mixture-of-Experts (MoE) models, we use the actual inference cost measured by:

$$cost = c_{in} * N_{in} + c_{out} * N_{out} \tag{1}$$

Where $c$ is the cost per token and $N$ is the number of tokens. We obtain the cost $c$ for the specific models a router chooses using the official API pricing published by the corresponding providers (e.g., OpenAI, Claude, Fireworks AI, etc.). For unpopular models that are not served by commercial providers, we deploy them ourselves for experiments and approximate their costs using the pricing tiers published by commercial hosting platforms (e.g., Together.ai), which estimate serving costs based on model size (parameter count) and architecture type (e.g., MoE) in Table 9.

**(3) Routing Optimality.** This captures a router's ability to perform optimal routing—that is, selecting the cheapest model that still produces a correct response. It consists of three sub-metrics: *(a) Optimal Selection Ratio*—the proportion of queries for which the router answers correctly by selecting the cheapest model; *(b) Optimal Accuracy Ratio*—the ratio between a router's achieved accuracy and the upper-bound accuracy obtainable when always selecting the best model from its model pool; *(c) Optimal Cost Ratio*—the ratio between the cost incurred by the router's selections and the cost of always choosing the optimal model. This metric will penalize routers that rely on unnecessarily expensive models when cheaper, correct alternatives are available.

**(4) Routing Robustness.** This metric evaluates the router's robustness against noisy inputs. We calculate it as the proportion of queries for which the router makes consistent routing decisions under perturbed input. Specifically, we generate noisy variants of queries—through paraphrasing, grammatical changes, synonym substitutions, and typos—and measure the percentage of cases where the router selects the same model as it does for the original, noise-free query. This captures the router's capability for handling realistic, imperfect user queries.

**(5) Routing Latency.** Since the router operates in the critical path of systems in production, it must handle millions of queries per second with minimal overhead. This metric measures the additional latency introduced by routing. It reflects the latency increase in both time-to-first-token (TTFT) and end-to-end response latency when a given router is employed.

## 5 ROUTERARENA EVALUATION FRAMEWORK

### 5.1 ARENA RANKING

ROUTERARENA provides a series of router leaderboards that enable users to compare the capabilities of different routers and select the best one suited to their scenarios. It includes six ranking scores based on the evaluation metrics described in Section 4, including Arena Score, Optimal-Selection-Ratio, Optimal-Acc-Ratio, Optimal-Cost-Ratio, Robustness, and Latency. Among these, the Arena Score $S_{i,\beta}$ captures the trade-off between accuracy and cost by combining them into a single composite measure using the Weighted Harmonic Mean (Ferger, 1931). Specifically, to better distinguish between routers with low costs, we apply a base-2 logarithmic ($log_2$) transformation to the cost values. Under this scaling, each doubling of price reduces the cost score by one unit. For router $i$ with cost $c_i$, we define its normalized cost as

$$C_i = \frac{log_2(c_{max}) - log_2 c_i}{log_2(c_{max}) - log_2(c_{min})} \tag{2}$$

where $c_{\max}$ and $c_{\min}$ denote the maximum and minimum costs of routing 1,000 queries. Specifically, we choose $c_{min} = 0.0044$, corresponding to the cost of the cheapest model in the leaderboard's model pool. This reflects the cost of a router that always selects the cheapest model. We set $c_{max} = 200$, which represents the most expensive model, OpenAI's O1-PRO. This normalization maps the cost into range $[0, 1]$, with larger values of $C_i$ corresponding to more economical routers. Next, we combine the normalized cost $C_i$ and accuracy $A_i$ using a weighted harmonic mean:

$$S_{i,\beta} = \frac{(1 + \beta)A_i C_i}{\beta A_i + C_i}, \tag{3}$$

where the parameter $\beta > 0$ controls the relative importance of accuracy versus cost. Intuitively, $\beta$ specifies the relative weighting applied to cost: we assign cost a weight of $\beta$ in comparison to accuracy. By default, we use $\beta = 0.1$, emphasizing routing accuracy, because highly accurate routers are generally more valuable even if they incur slightly higher costs.

Details about how we update $c_{min}$ and $c_{max}$, recommendations on values of $\beta$ for different user needs are provided in Appendix H.

### 5.2 AUTOMATED EVALUATION FRAMEWORK

Although we demonstrate ROUTERARENA with a specific set of routers in this paper, it is very easy to update the leaderboard with new routers. To facilitate this process, we have designed an automated evaluation framework that has been released publicly alongside the leaderboard. Figure 5 shows the overall system workflow. To evaluate a new router, the user can simply start by providing our framework with an access point (e.g., an API) to the router. The framework sends evaluation queries to the router, which performs routing inference on its end and returns its model selections. To ensure fairness, we run the inference ourselves and use cached results when possible, since many routers share overlapped model pools. During this process, the router's response time is monitored to measure routing latency. Finally, the framework computes the evaluation metrics and aggregates the results, which are reflected in the leaderboard. Note that some commercial routers may not expose their model selections and instead return query answers directly. Such routers can still be evaluated with our framework, although certain metrics cannot be measured in this setting.

## 6 EXPERIMENTS

### 6.1 EXPERIMENTAL SETTINGS

**Router Selection.** For commercial routers, we evaluated the router from Not Diamond (NotDiamond, 2025), which provides access to over 60 models, and the Azure Model Router (Microsoft,

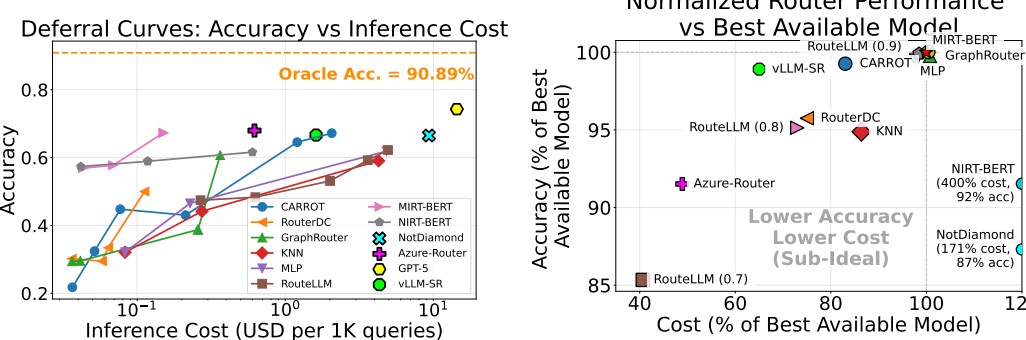

Figure 5: The automated router evaluation and leaderboard update framework.

2025), which currently only supports OpenAI models. We also included GPT-5 (OpenAI, 2025), whose model family incorporates an internal router. For NotDiamond, we selected 26 representative models spanning different parameter scales, architectures, and reasoning abilities. For Azure-Router, we evaluated the entire model pool, including GPT-5 model families. Appendix B provides the full details of the model pools used for each router.

For open-source routers, we evaluated nine representative systems covering a diverse routing approaches. Specifically, we chose both the KNN- and MLP-based methods trained on Router-Bench (Hu et al., 2024) as baselines. We further included GraphRouter (Feng et al., 2025a), which leverages graph neural networks (GNNs) for routing, and the Universal Router (Jitkrittum et al., 2025), which uses K-means clustering. To capture cost-accuracy tradeoffs, we evaluated CARROT Router (Somerstep et al., 2025), while RouterDC (Chen et al., 2024) was incorporated as a dual contrastive learning-based approach. Additionally, we considered IRT-Router (Song et al., 2025), which applies item response theory to explicitly model the interaction between query attributes and model capabilities, and RouteLLM (Ong et al., 2025), which performs binary selection between a stronger and a weaker model. Moreover, we also take the latest vLLM Semantic Router (vLLM, 2025) into consideration, which leverages ModernBERT (Hugging Face, 2025) to categorize the incoming requests into pre-defined categories, and selects the model that has the highest score.

**Router Training and Evaluation.** For commercial routers, no additional training is required; we simply accessed their provided APIs for evaluation. In contrast, for academia routers, we followed the training procedures and datasets specified in their open-source implementations. Specifically, we did not modify the training datasets or the task categorizations (if applicable). The model pools were configured in accordance with the original papers. In particular, for the vLLM-SR, we constructed the pool using both open-source models of varying parameter scales and proprietary models, with detailed configurations summarized in Table 3. After training each router, we evaluated them by feeding our benchmark dataset, recording the model selected, the latency incurred by the selection, and the confidence scores assigned to all candidate models in the pool.

## 6.2 RESULTS

Figure 6: Deferral Curve: accuracy versus cost

Figure 7: Normalized Scatter Plot

**Deferral Curve.** Figure 6 presents the trade-off between accuracy and inference cost. As we increase inference budget, we unlock more powerful models, driving the accuracy up. For open-source routers, we leveraged their confidence scores to apply budget-based masking, which produces multiple points along each curve. With only cheap models available, accuracy remains low, but as larger models enter the pool, routing accuracy increases. In contrast, commercial routers typically appear as single points because their model pools already include the best-performing models.

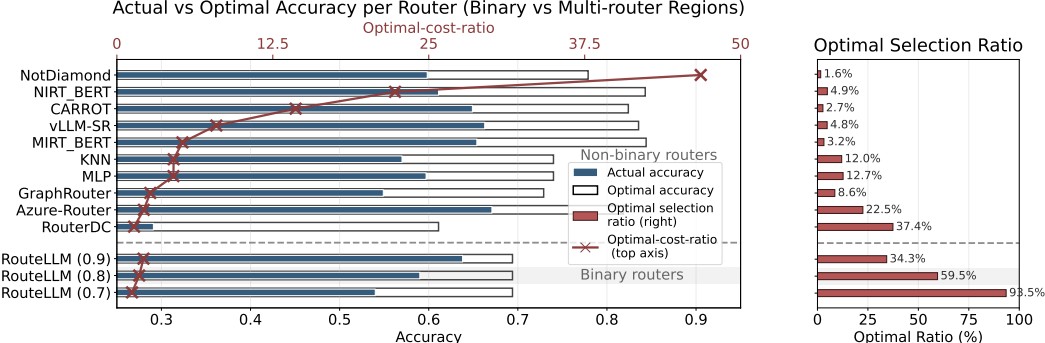

Figure 8: Actual and optimal accuracy, along with optimal selection ratio and cost ratio

Two insights emerge. First, the orange dashed line shows the oracle accuracy, highlighting that all routers fall short of the best achievable performance. Second, the trade-off frontier differs by setting: commercial routers can achieve higher accuracy, but usually at significantly higher costs; open-source routers, on the other hand, achieve competitive performance at much lower budgets, though they plateau earlier. Notably, routers like CARROT and GraphRouter illustrate cost-efficient routing, while systems such as GPT-5 and NotDiamond lean heavily on expensive models for accuracy. This suggests that while commercial routers prioritize maximizing accuracy, academic approaches often explore the efficiency side of the frontier.

**Normalized Deferral Curve.** Figure 7 reports router accuracy and cost normalized to each router's best-performing model, point (100%, 100%) on the plot. The upper-left quadrant represents the ideal case—higher accuracy with lower cost by leveraging smaller models. In practice, most routers cluster near the baseline (100% cost, 100% accuracy), suggesting they over-rely on the strongest model and miss opportunities to defer to cheaper alternatives. Notably, NIRT-BERT illustrates inefficiency, reaching only baseline-level accuracy while incurring 400% of the cost.

By contrast, routers such as vLLM-SR and CARROT achieve meaningful savings: roughly 35% lower cost with under 2% accuracy degradation. These cases show routing can indeed improve efficiency when smaller models are effectively utilized. Overall, the results highlight a clear trade-off—higher accuracy often comes with higher cost—while also pointing to promising directions for designing routers that move closer to the ideal frontier.

**Optimality Score.** Figure 8 highlights the inherent trade-off between routing accuracy and cost. In practice, routers that achieve higher accuracy typically do so at the expense of a higher cost ratio, since they defer more often to large, expensive models. This behavior lowers their optimal selection ratio, i.e., the frequency with which they choose the most efficient model for each query. This pattern is most apparent in the binary routers such as RouteLLM. By design, these routers face a sharp trade-off: they achieve higher accuracy by routing more queries to the stronger model, which drives up cost. In contrast, multi-model routers have a more flexible pool, and while the general trend still holds, we see greater variability depending on pool composition and routing strategy. Among non-binary routers, RouterDC stands out with the lowest cost ratio and highest optimal selection ratio, but this comes at the cost of poor overall accuracy. At the other extreme, MIRT-BERT achieves strong accuracy (close to 77% of its optimal accuracy) but requires nearly five times the optimal cost, placing it closer to the "high-cost high-accuracy" region of the trade-off frontier. In other words, while some routers are closer to the efficiency frontier than others, none simultaneously combine low cost and high accuracy. Overall, our findings indicate that current routing methods have learned to leverage large models to boost performance, but remain inefficient at recognizing when smaller models are sufficient. This creates a clear opportunity for future work: developing routers that can balance accuracy and efficiency by selectively deferring to large models only when necessary.

**Robustness and Latency.** Given that user prompts are often noisy, we further assess routing sensitivity and robustness. The detailed evaluation is in Appendix G. We define robustness as 1 − the proportion of changed selections. As shown in Figure 9 (left), robustness is uniformly low across both commercial routers and academic routers such as vLLM-SR, MIRT-BERT, and NIRT-BERT. A key reason is that these academic routers depend on BERT to compute latent representations,

and BERT is known to be sensitive to surface-level perturbations (Jin et al., 2020). These findings highlight the importance of applying prompt engineering techniques to mitigate noisy queries.

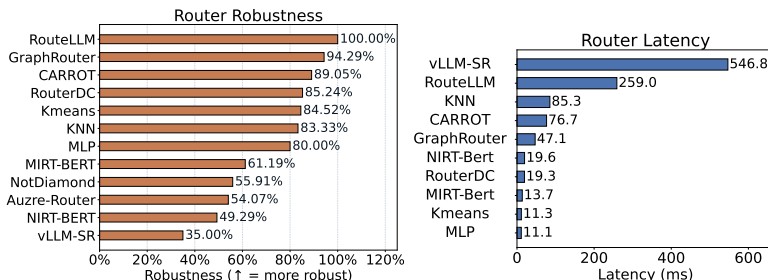

Figure 9: Router robustness and latency comparison.

Furthermore, Figure 9 (right) reports the end-to-end latency of routers on a single A100 GPU, measured from the time a request is received to the output of the model selection result. Among all methods, vLLM-SR and RouteLLM exhibit significantly higher latency because they rely on the OpenAI embedding API, which introduces additional network delays. In contrast, others consistently maintain sub-100ms latencies. While LLM routers can optimize accuracy and cost, they also introduce non-negligible overhead that may even compromise service-level objectives (SLOs).

**Generation Length** We also report the average generation length and price per M output tokens for each router in Table 8. GPT-5 and Azure-Router rely more heavily on reasoning models, leading to substantially longer generations than most other routers. Longer answers are not inherently better; this metric simply offers an additional axis for users to trade off concise versus detailed responses.

### 6.3 RESULTS BY DIFFICULTY LEVELS

Table 6 shows the accuracy-cost tradeoff across easy, medium, and hard queries. Most routers exceed 89% accuracy on easy questions, but accuracy drops sharply on medium and especially hard ones (often below 10%), confirming that our difficulty bands are non-trivial. GPT-5 achieves the highest accuracy at every level but at a much higher cost. At the same time, routers such as Azure-Router and vLLM-SR offer competitive accuracy on easy/medium queries at lower cost. This contrast reveals substantial headroom on hard questions and clear efficiency-accuracy tradeoffs across methods. We also observe that some routers (e.g., GPT-5, Azure-Router, MIRT-BERT, NotDiamond) allocate significantly more budget to hard than easy queries, while others show almost flat cost across difficulty, suggesting room for more difficulty-aware routing strategies.

### 6.4 ADDITIONAL RESULTS ON LONGBENCH-V2

We provide additional results on LongBench-v2 in Appendix E. On this long-context suite, GPT-5 attains the highest accuracy, while Azure-Router and NIRT-BERT offer competitive performance at substantially lower cost, illustrating clear accuracy-cost trade-offs in the long-context regime.

### 6.5 INSIGHTS FROM ROUTERARENA'S LEADERBOARD

Our evaluation produces the router leaderboard shown in Table 2. The leaderboard consists of six ranking scores, and the overall ranking is determined by averaging across them. We highlight two key findings: (1) Commercial routers do not necessarily outperform open-source routers. For example, GPT-5 ranks #7 due to its restricted model pool, and NotDiamond ranks #12 because it frequently selects expensive models. (2) No router ranks at the top across all metrics, reflecting the inherent trade-offs in router design.

For developers and researchers, the findings highlight key deficiencies in current routing methods and point toward clear directions for designing the next generation of routers. The results show that all existing routers fall short of the oracle's achievable performance, primarily because they are inefficient at recognizing when smaller, cheaper models are sufficient for a given query. Future work should focus on closing this performance gap. Moreover, the high latency and poor robustness of

Table 2: Ranking of routers across multiple metrics. Lower values indicate better performance.

| # | Router | Arena Rank | Optimal-Selection Ratio | Optimal-Cost Ratio | Optimal-Acc Ratio | Robustness Rank | Latency Rank | Average |
|---|--------|-----------|------------------------|--------------------|-------------------|-----------------|--------------|---------|
| 1 | Azure-Router | 2 | 3 | 3 | 5 | 9 | - | 4.40 |
| 2 | MIRT-BERT | 1 | 9 | 7 | 1 | 7 | 2 | 4.50 |
| 3 | RouteLLM | 9 | 1 | 1 | 10 | 1 | 8 | 5.0 |
| 4 | GPT-5 | 5 | - | - | - | - | - | 5.0 |
| 5 | GraphRouter | 7 | 6 | 4 | 9 | 2 | 5 | 5.50 |
| 6 | MLP | 10 | 4 | 5 | 8 | 6 | 1 | 5.67 |
| 7 | Router DC | 12 | 2 | 2 | 11 | 4 | 3 | 5.67 |
| 8 | CARROT | 3 | 10 | 9 | 4 | 3 | 6 | 5.83 |
| 9 | NIRT-BERT | 6 | 7 | 10 | 2 | 10 | 4 | 6.5 |
| 10 | KNN | 11 | 5 | 6 | 7 | 5 | 7 | 6.83 |
| 11 | VLLM-SR | 4 | 8 | 8 | 3 | 11 | 9 | 7.17 |
| 12 | NotDiamond | 8 | 11 | 11 | 6 | 8 | - | 8.80 |

certain routers open new avenues of research beyond the traditional cost-accuracy trade-off. Developers can use the platform's automated framework to submit and benchmark new routers against established leaders, fostering innovation and transparently tracking progress in the field.

# 7 RELATED WORK

**LLM Router.** With the increasing availability of specialized models that can surpass even the most capable general-purpose LLMs in specific domains, both academia and industry have been actively exploring how to build LLM routers. In industry, several systems have emerged. Martian Router (WithMartian, 2025) proposed the idea of model mapping, while Storytell (Storytell.ai, 2025) categorizes user queries and routes them to the best-performing models. Other companies also seek to find the optimal model for user's tasks by balancing performance and cost (NotDiamond, 2025; RequestyAI, 2025; OpenAI, 2025). Recent academic efforts have also begun to emerge. GraphRouter (Feng et al., 2025a) leverages graph neural networks, and Router-R1 (Zhang et al., 2025a) employs reinforcement learning. The growth of open-source solutions underscores the need for effective router evaluation (Somerstep et al., 2025; Song et al., 2025; Feng et al., 2025b).

**LLM Router Benchmarks.** RouterBench (Hu et al., 2024) introduces a large-scale dataset consisting of over 405k inference outcomes from representative LLMs. RouterEval (Huang et al., 2025) collects performance results from 8,500 LLMs across 12 widely used benchmarks. Fusion-Bench (Feng et al., 2025b) covers 14 tasks across five domains and leverages 20 open-source LLMs. Other benchmarks have also contributed to this line of work by using different data collection methods (Kassem et al., 2025; Mei et al., 2025). All of these benchmarks adopt an *offline* design: they release a large frozen corpus of model outputs that is ideal for cheap, fully reproducible comparison of routing algorithms under a fixed model pool. By contrast, RouterArena is explicitly designed as a *live leaderboard* that includes both academic and commercial routers. Our live leaderboard by nature encourages router designers to explore evolving and specialized models, making routing decisions more meaningful in practice. Meanwhile, RouterArena improves on prior router benchmarks by covering more tasks and domains, and by providing a dataset with empirically calibrated difficulty together with more comprehensive metrics.

# 8 CONCLUSION

We introduce ROUTERARENA, the first open platform for comprehensive router comparison. Our platform features a principled dataset with broad domain coverage and varying difficulty levels, an extensive set of evaluation metrics, and an automated framework to maintain a live leaderboard. Initial evaluations of 12 routers reveal a significant trade-off between accuracy and cost, showing that no single router is universally optimal. Commercial routers tend to achieve higher accuracy at a much greater expense, while open-source routers often present more cost-efficient solutions. Overall, our findings indicate that current routers are inefficient at leveraging cheaper models when appropriate, highlighting a clear opportunity for future work.

ETHICS STATEMENT

All authors have read and will adhere to the ICLR Code of Ethics (`https://iclr.cc/public/CodeOfEthics`). Our evaluation dataset was constructed by aggregating and sampling from publicly available and open-source datasets. To ensure broad topic coverage while avoiding potentially sensitive subjects, we based our domain selection on the Dewey Decimal Classification system, explicitly excluding the category of religion. The difficulty level of each query was annotated using an "LLM-as-Judge" approach. We acknowledge that this process, along with the use of existing datasets, may introduce or perpetuate biases inherent in the source data and the annotator model. We have made the dataset and our collection methodology public to allow for further inspection and bias analysis by the community.

REPRODUCIBILITY STATEMENT

We are committed to ensuring the full reproducibility of our work. All components of the ROUTER-ARENA platform, including the dataset, evaluation framework, and code, will be made publicly available.

(1) Dataset: The principles and detailed process for our dataset construction are described in Section 3. This includes domain coverage strategy, difficulty level annotation, and our deduplication process. Further details on the dataset schema, composition, and examples are provided in Appendix C. (2) Evaluation: Our five evaluation metrics are precisely defined in Section 4, and the formula for the composite leaderboard score is detailed in Section 5.1. The automated evaluation framework is described in Section 5.2. (3) Experiments: The specific academic and commercial routers evaluated are listed in Section 6.1. The exact model pools used for each router are provided in Appendix B. For all academic routers, we followed the training procedures and configurations specified in their original open-source implementations.

The public release of our complete framework will enable researchers to replicate our results and evaluate new routers on the leaderboard.

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

## A  THE USE OF LLMS

In this work, Large Language Models (LLMs) were used in two distinct capacities: as a core component of our research methodology and as a general-purpose writing aid.

LLM-as-Judge for Data Annotation: A significant use of an LLM was in the construction of our evaluation dataset. As detailed in Section 3, we employed DeepSeek-V3.1 as an "LLM-as-Judge" to automatically annotate the cognitive level of each query according to Bloom's taxonomy. This automated approach allowed us to systematically label a large and diverse set of questions. The specific prompt used for this annotation process is provided in Appendix D.

Writing Assistance: An LLM was also utilized as a general-purpose tool to assist in the writing process. Its role was limited to proofreading the manuscript for grammatical errors, improving clarity, and ensuring stylistic consistency. The LLM was not used for research ideation, conducting experiments, or writing the core scientific contributions of the paper.

## B  MODEL POOLS BY ROUTER

Table 3: Model pools used by different routers.

| Router | Model Pool |
|---|---|
| RouterBench | WizardLM/WizardLM-13B-V1.2; claude-instant-v1; claude-v1; claude-v2; gpt-3.5-turbo-1106; gpt-4-1106-preview; meta/codellama-34b-instruct; meta/llama-2-70b-chat; mistralai/mistral-7b-chat; mistralai/mixtral-8x7b-chat; zero-one-ai/Yi-34B-Chat |
| GraphRouter | meta-llama/llama-3-8b-instruct; mistralai/mixtral-8x7b-chat; nousresearch/nous-34b-chat; meta/llama-2-7b-chat; mistralai/mistral-7b-chat; meta/llama-3-70b-chat; meta/llama-3-turbo-8b-chat; meta/llama-3-turbo-70b-chat; meta/llama-3.1-turbo-70b-chat; qwen/qwen-1.5-72b-chat |
| Universal | WizardLM/WizardLM-13B-V1.2; claude-instant-v1; claude-v1; claude-v2; gpt-3.5-turbo-1106; gpt-4-1106-preview; meta/codellama-34b-instruct; meta/llama-2-70b-chat; mistralai/mistral-7b-chat; mistralai/mixtral-8x7b-chat; zero-one-ai/Yi-34B-Chat |
| CarrotRouter | aws-claude-3-5-sonnet-v1; aws-titan-text-premier-v1; openai-gpt-4o; openai-gpt-4o-mini; wxai-granite-3-2b-instruct-8k-max-tokens; wxai-granite-3-8b-instruct-8k-max-tokens; wxai-llama-3-1-70b-instruct; wxai-llama-3-1-8b-instruct; wxai-llama-3-2-1b-instruct; wxai-llama-3-2-3b-instruct; wxai-llama-3-3-70b-instruct; wxai-mixtral-8x7b-instruct-v01; wxai-llama-3-405b-instruct |
| RouterDC | mistralai/Mistral-7B-v0.1; meta-math/MetaMath-Mistral-7B; itpossible/Chinese-Mistral-7B-v0.1; HuggingFaceH4/zephyr-7b-beta; cognitivecomputations/dolphin-2.6-mistral-7b; meta-llama/llama-3-8b-instruct; cognitivecomputations/dolphin-2.9-llama3-8b |
| IRT-Router | glm_4_air; glm_4_flash; glm_4_plus; gpt_4o; gpt_4o_mini; gpt_4o_mini_cot; deepseek_coder; deepseek_chat; qwen25_32b_int4; qwen25_7b_instruct; qwen25_72b_instruct; qwq_32b_preview; qwen25_math_7b_instruct; llama31_8b_instruct; llama31_70b_instruct; llama31_405b_instruct; mixtral_8x7b_instruct; mistral_7b_instruct_v02; ministral_8b_instruct_2410; gemini15_flash; claude35_haiku20241022 |
| RouteLLM | openai-gpt-4o; mixtral_8x7b_instruct |

We provide the model pool used by each router here.

## C  DATASETS DETAILS

### C.1  USE OF LLM-AS-A-JUDGE

To determine the alignment of LLM as a judge with human judgment of Bloom's level of the query, we sampled 450 queries from the dataset by stratifying along both the domain and Bloom level, ensuring that each combination was proportionally represented. Eighteen volunteers (undergraduate and graduate students) were asked to assign a Bloom level to each query under the same instructions as used for the DEEPSEEK-V3.1 judge. For each query, we compare the human label with

the LLM's prediction. We observe 54.9% exact agreement and 76.7% agreement within $\pm 1$ Bloom level, indicating that most disagreements occur between adjacent levels in this inherently subjective taxonomy. Since Bloom labels are used only to ensure cognitive-skill coverage during dataset construction—while all evaluation difficulty is defined empirically via accuracy over 42 LLMs—this level of agreement is sufficient for our purposes.

## C.2 DATASET COMPOSITION AND SCHEMA

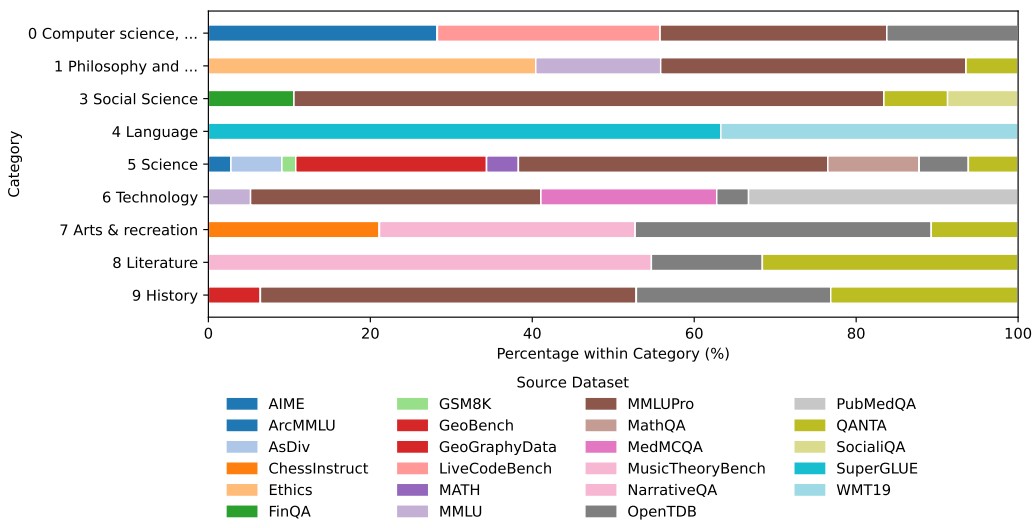

Figure 10: The source of ROUTERARENA evaluation dataset.

Figure 10 illustrates the domain coverage of our dataset across the nine Dewey Decimal categories. Each horizontal bar represents the relative contribution of different source datasets within a category. This distribution highlights the systematic integration of multiple datasets to achieve a more balanced representation of both general-purpose and highly specialized domains.

Table 4: Overview of dataset columns

| Column | Description | Example |
|---|---|---|
| Category | Bloom's taxonomy high-level class | 9 History |
| Sub Category | Bloom's taxonomy sub-class | 02 Library and information sciences |
| Dataset Name | Source dataset | ArcMMLU |
| Global Index | Unique instance ID | ArcMMLU_114 |
| Context | Supporting passage (if any) | Sasha decided to watch TV and get some food |
| Question | Input question | What is the capital of France? |
| Options | Multiple-choice options | ["10", "20", "30", "40"] |
| Answer | Ground-truth answer | Paris |
| Bloom Level | Bloom's taxonomy difficulty level | Understanding |

Table 4 shows the detailed dataset columns.

## C.3 DATASET EXAMPLES

Figure 11 shows some dataset examples.

Table 5: The 42 models used for empirical difficulty labeling. Models span across a range of sizes and performances, showcasing that ROUTERARENA could distinguish LLMs by providing diverse questions of difficulty.

| Rank | Model | Score [%] |
|---|---|---|
| 1 | gpt-5 | 74.04 |
| 2 | claude-3-5-sonnet | 67.73 |
| 3 | deepseek/deepseek-chat | 66.86 |
| 4 | gpt-4o | 66.35 |
| 5 | llama-3-1-405b-instruct | 65.43 |
| 6 | glm-4-plus | 62.02 |
| 7 | gpt-4-1106-preview | 61.91 |
| 8 | meta-llama_Meta-Llama-3.1-70B-Instruct-Turbo | 60.64 |
| 9 | Qwen/Qwen2.5-32B-Instruct-GPTQ-Int4 | 60.51 |
| 10 | Qwen/Qwen2.5-72B-Instruct | 59.98 |
| 11 | meta-llama_Llama-3.1-70B-Instruct | 59.89 |
| 12 | Qwen/QwQ-32B | 59.81 |
| 13 | gpt-4o-mini | 59.22 |
| 14 | meta-llama_Meta-Llama-3-70B-Instruct | 58.16 |
| 15 | google/gemini-flash-1.5 | 57.21 |
| 16 | glm-4-air | 54.63 |
| 17 | Qwen/Qwen1.5-72B-Chat | 49.86 |
| 18 | Qwen/Qwen2.5-7B-Instruct | 47.70 |
| 19 | glm-4-flash | 47.62 |
| 20 | mistralai/Mixtral-8x7B-Instruct-v0.1 | 47.01 |
| 21 | 01-ai/Yi-34B-Chat | 45.54 |
| 22 | llama-3-1-8b-instruct | 45.23 |
| 23 | NousResearch/Nous-Hermes-2-Yi-34B | 45.01 |
| 24 | meta-llama_Llama-3.1-8B-Instruct | 44.24 |
| 25 | meta-llama_Llama-2-70b-chat-hf | 39.82 |
| 26 | meta/llama-2-70b-chat | 39.49 |
| 27 | mistralai/Ministral-8B-Instruct-2410 | 32.87 |
| 28 | llama-3-2-3b-instruct | 32.55 |
| 29 | cognitivecomputations/dolphin-2.9-llama3-8b | 32.35 |
| 30 | Qwen/Qwen2.5-32B | 32.26 |
| 31 | mistralai/Mistral-7B-Instruct-v0.2 | 31.83 |
| 32 | meta-llama_llama-3-8b-instruct | 30.45 |
| 33 | cognitivecomputations/dolphin-2.6-mistral-7b | 29.98 |
| 34 | mistralai/Mistral-7B-Instruct-v0.3 | 28.05 |
| 35 | meta-llama_Meta-Llama-3-8B-Instruct | 23.69 |
| 36 | WizardLM/WizardLM-13B-V1.2 | 23.03 |
| 37 | llama-3-2-1b-instruct | 22.10 |
| 38 | meta-llama_Llama-2-7b-chat-hf | 20.50 |
| 39 | meta/codellama-34b-instruct | 20.42 |
| 40 | codellama/CodeLlama-34b-Instruct-hf | 19.31 |
| 41 | meta-llama_Meta-Llama-2-7B | 18.92 |
| 42 | meta-llama_Meta-Llama-2-7B-hf | 18.39 |

Figure 11: Dataset examples.

## C.4 MODEL LIST FOR EMPIRICAL DIFFICULTY LABELING

We use the following 42 models to compute the empirical difficulty labels shown in Table 5. These models are drawn from the model pools of the routers we benchmark, and were chosen to cover a broad range of parameter scales, architectures, and providers. This diversity ensures that queries elicit a wide spread of accuracies across models, which we indeed observe in their scores and use as the basis for our empirical difficulty measure.

# D LLM PROMPTS

## D.1 BLOOM TAXONOMY PROMPT

We used the following prompt for classifying each question into different Bloom Levels:

---

**Instruction:** You are an evaluator tasked with classifying questions by cognitive difficulty using **Bloom's Taxonomy (Revised 2001)**. Bloom's Taxonomy defines six levels of cognitive processes:

1. **Remember** – Recall or recognize facts, terms, or concepts. 2. **Understand** – Explain, summarize, interpret, or demonstrate comprehension. 3. **Apply** – Use learned knowledge to solve problems in new or routine situations. 4. **Analyze** – Break down information, examine parts, relationships, or underlying causes. 5. **Evaluate** – Make judgments or decisions based on evidence, criteria, or standards. 6. **Create** – Put elements together to form a new structure, idea, or product.

**Your Task:** Given the details of a question, determine *which Bloom's level best represents the cognitive process required to answer it*.

- Output the result in a **structured JSON block** with no additional text. - If multiple levels might apply, choose the *highest* level required.

**Output Format:**

```
{
  "bloom_level": "<Remember | Understand | Apply | Analyze |
  Evaluate | Create>"
}
```

**Input Fields:**

```
– Dataset name: {dataset_name}
– Dataset classification (DDC category): {dataset_category}
– Question: {question}
– Question Index: {question_index}
– Context (if any): {context}
– Options (if MCQ): {options}
– Answer: {answer}
```

---

Table 6: Router performance by difficulty level (accuracy and cost per 1k queries).

| Router | Overall | Easy | Medium | Hard |
|---|---|---|---|---|
| GPT-5 | 74.0% ($14.02) | 95.1% ($5.68) | 68.6% ($14.80) | 27.5% ($35.73) |
| Azure-Router | 68.1% ($0.54) | 93.3% ($0.30) | 59.5% ($0.63) | 17.9% ($1.05) |
| NotDiamond | 68.0% ($9.34) | 92.5% ($6.34) | 61.9% ($10.29) | 15.1% ($15.16) |
| vLLM-SR | 67.3% ($1.67) | 95.3% ($1.56) | 57.9% ($1.70) | 8.7% ($1.87) |
| CARROT | 67.2% ($2.06) | 95.0% ($1.79) | 58.0% ($2.26) | 9.0% ($2.36) |
| MIRT-BERT | 66.9% ($0.15) | 95.9% ($0.10) | 56.8% ($0.17) | 7.1% ($0.26) |
| NIRT-BERT | 62.0% ($0.69) | 93.7% ($0.53) | 47.0% ($0.78) | 4.7% ($0.90) |
| MLP | 61.6% ($4.83) | 93.2% ($3.54) | 46.6% ($5.74) | 4.6% ($6.52) |
| KNN | 58.7% ($4.27) | 89.2% ($3.08) | 43.6% ($5.13) | 4.5% ($5.77) |
| GraphRouter | 57.0% ($0.34) | 89.7% ($0.25) | 39.3% ($0.39) | 2.4% ($0.46) |
| RouteLLM | 47.0% ($0.27) | 79.8% ($0.20) | 24.4% ($0.33) | 2.0% ($0.32) |
| RouterDC | 32.0% ($0.07) | 54.9% ($0.06) | 15.3% ($0.08) | 2.2% ($0.09) |

Table 7: RouterArena routers—long-context performance and cost. RouteLLM cannot be evaluated on this subset because its text encoder only supports inputs up to 8,192 tokens.

| Router | Acc (%) | Cost / 1K Queries (USD) |
|---|---|---|
| GPT-5 | 71 | 45.70 |
| NotDiamond | 70 | 59.30 |
| Azure-Router | 67 | 9.54 |
| NIRT-BERT | 60 | 6.60 |
| RouteLLM | – | – |
| GraphRouter | 42 | 23.60 |
| vLLM-SR | 42 | 64.80 |
| CARROT | 25 | 4.10 |
| RouterDC | 12 | 3.60 |
| MIRT-BERT | 60 | 7.50 |
| MLP | 65 | 286.70 |
| KNN | 69 | 300.30 |

## D.2 EVALUATION PROMPT

The following is an example of an evaluation prompt we use:

```
Please read the following multiple-choice questions and
consider which option best follows deontological ethics.\n\n
Context: {Context}\n\n
Question: {Question}\n\n
Options: \n{Options}\n\n
Provide the correct letter choice in \\boxed{{X}}, where X is
the correct letter choice.
Keep the explanation or feedback within 3 sentences.
```

## E ADDITIONAL RESULTS AND TABLES:

**Long-Context Routing.** Table 7 reports router performance on the LongBench-v2 dataset. We randomly sampled 100 queries of short and medium length from the dataset. GPT-5 attains the highest accuracy (71%) but also the highest cost per 1k queries. Azure-Router, MIRT-BERT, and NIRT-BERT achieve reasonably strong accuracy (60–67%) at one order of magnitude lower cost, whereas methods like MLP, KNN, NotDiamond, and vLLM-SR are substantially more expensive for comparable or worse accuracy. RouteLLM cannot be evaluated in this setting because its text encoder only supports inputs up to 8,192 tokens, which excludes many of our long-context queries.

**Generation Length and Cost.** Table 8 summarizes average generation length and cost per million output tokens. GPT-5 produces the longest responses and is also the most expensive per token,

Table 8: RouterArena routers—average generation length and cost.

| Router | Avg. Gen. Length | Cost / M Tokens (USD) |
|---|---|---|
| GPT-5 | 970.65 | 10.00 |
| NotDiamond | 390.05 | 9.18 |
| Azure-Router | 270.90 | 1.79 |
| NIRT-BERT | 215.82 | 1.33 |
| RouteLLM | 140.24 | 0.68 |
| GraphRouter | 132.24 | 0.81 |
| vLLM-SR | 109.27 | 9.03 |
| CARROT | 105.56 | 12.58 |
| RouterDC | 102.83 | 0.20 |
| MIRT-BERT | 89.16 | 1.00 |
| MLP | 81.06 | 29.75 |
| KNN | 79.80 | 26.31 |

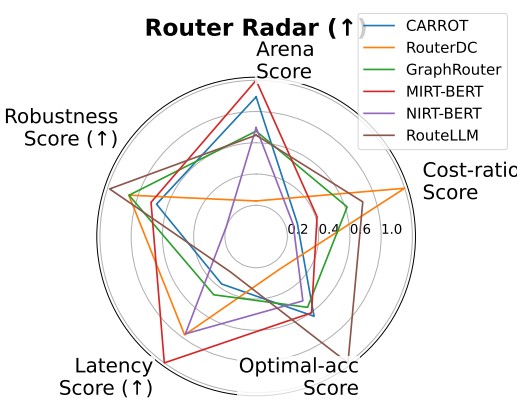

Figure 12: The spider plot of ROUTERARENA

while Azure-Router, NIRT-BERT, and MIRT-BERT generate shorter outputs at much lower cost. In contrast, heuristic baselines such as MLP, KNN, and CARROT have relatively short generations yet very high token-level costs, indicating that they are inefficient choices when generation cost is a primary concern.

## F   LEADERBOARD

Figure 12 presents the spider plot of RouterArena, which compares six routing methods (CARROT, RouterDC, GraphRouter, MIRT-BERT, NIRT-BERT, and RouteLLM) across five evaluation dimensions: Arena Score, Cost-ratio Score, Optimal-acc Score, Latency Score, and Robustness Score. Each axis indicates higher performance in the outward direction, allowing a direct visualization of trade-offs. For example, CARROT achieves strong performance in Arena and Latency Scores, while RouterDC excels in Cost-ratio Score.

## G   ROBUSTNESS EVALUATION

Specifically, we assign equal weight to the four noisy variants described in the definition. For each category and its subcategories, we randomly sampled 5% of the entire dataset, which is 420 examples and generated perturbed queries using GPT-4o. The exact prompt we used to generate perturbed queries is as follows:

You are an assistant that performs extremely strong and highly disruptive text transformations for robustness stress testing. You must follow the selected mode exactly. Transformations must be intense, obvious, and aggressive. Always preserve the requested structure, but you may heavily distort surface wording. Do NOT shorten or expand the text beyond the allowed limits.

There are four transformation modes:

PARAPHRASE (MAXIMAL TRANSFORMATION MODE) Rewrite the text so heavily that almost no original sentences or phrasing remain. You may reorder ideas, split or merge sentences, and aggressively substitute expressions. The meaning must stay intact, but the wording should look almost unrecognizable from the input. Keep length approximately the same (±15

GRAMMATICAL RECONSTRUCTION (DEEP REWRITE MODE) Rebuild the text with fully restructured, polished, and elegant grammar. You may reorder clauses, shift emphasis, modernize tone, and transform the writing style completely. All information must remain, but the writing should feel dramatically improved and profoundly different from the original. Keep the length the same (±10

SYNONYM SATURATION (ULTRA-DENSE SUBSTITUTION MODE) Replace nearly every content word (nouns, verbs, adjectives, adverbs) with strong, less common, or surprising synonyms. You may alter idioms, swap multiword phrases with equivalent expressions, and push semantic boundaries while preserving meaning. The output should visually look almost entirely changed. Maintain length (±10

INTENTIONAL CORRUPTION (HEAVY DEGRADATION MODE) Inject a very high density of diverse, realistic, and chaotic textual corruptions: misspellings, keyboard slips, letter swaps, missing characters, inserted random characters, spacing glitches, partial word breakage, and inconsistent capitalization. Meaning must remain barely decipherable but still recoverable. Maintain approximate length (±10

Input format:

MODE: <Paraphrase — Grammar — Synonyms — Typos>

TEXT: <user_provided_text>

Output format (strict):

=== TRANSFORMED TEXT === <extremely modified text>

=== NOTES === <1–2 very short descriptions of the corruption intensity and strategy>

Do not output anything else.

## H    DISCUSSION ABOUT HYPERPARAMETER($c_{min}, c_{max}, \beta$)

In our framework, $c_{max}$ and $c_{min}$ are not intended to be static global constants; instead, they are tunable for flexibility. To prevent the most expensive router from being normalized to zero, for now, we set $c_{max}$ and $c_{min}$ to the prices of the most expensive and cheapest models on the market. However, whenever a model's actual cost really falls below $c_{min}$ or exceeds $c_{max}$, the only thing we need to do is to adjust the $c_{min}$ and $c_{max}$, and re-calculate the Arena Score so that all normalized costs remain within $(0, 1)$. Note that this does not require re-running the experiment.

The parameter $\beta$ is also adjustable to modulate how much weight is assigned to cost compared to accuracy. Specifically, we made the $\beta$ tunable on our leaderboard for users. Here we give three example settings:

- Example 1: $\beta = 0.01$, Accuracy : Cost = 100, accuracy-dominated ranking for users who do not consider cost to be a key constraint.

- Example 2: $\beta = 0.1$, Accuracy : Cost = 10, trade-off balanced ranking for users who primarily care about accuracy but still operate under budget constraints.

- Example 3: $\beta = 1$, Accuracy : Cost = 1, equally weighted ranking for users who have a limited budget.

| Router | Rank & Score @ β=0.01 | Rank & Score @ β=0.1 | Rank & Score @ β=1 | Accuracy | Norm.Cost |
|---|---|---|---|---|---|
| GPT-5 | 1 (0.7282) | 5 (0.6272) | 12 (0.3688) | 0.7428 | 0.2453 |
| Azure-Router | 2 (0.6780) | 2 (0.6640) | 2 (0.6010) | 0.6798 | 0.5386 |
| MIRT-BERT | 3 (0.6731) | 1 (0.6729) | 1 (0.6718) | 0.6731 | 0.6705 |
| CARROT | 4 (0.6682) | 3 (0.6386) | 6 (0.5219) | 0.6720 | 0.4266 |
| vLLM-SR | 5 (0.6633) | 4 (0.6385) | 5 (0.5368) | 0.6665 | 0.4494 |
| NotDiamond | 6 (0.6565) | 8 (0.5935) | 11 (0.3998) | 0.6651 | 0.2858 |
| RouteLLM | 7 (0.6175) | 9 (0.5800) | 10 (0.4440) | 0.6224 | 0.3451 |
| NIRT-BERT | 8 (0.6151) | 6 (0.6083) | 4 (0.5764) | 0.6159 | 0.5416 |
| MLP | 9 (0.6143) | 10 (0.5780) | 9 (0.4449) | 0.6191 | 0.3472 |
| GraphRouter | 10 (0.6070) | 7 (0.6054) | 3 (0.5976) | 0.6072 | 0.5884 |
| KNN | 11 (0.5867) | 11 (0.5577) | 8 (0.4464) | 0.5905 | 0.3588 |
| RouterDC | 12 (0.3362) | 12 (0.3522) | 7 (0.4627) | 0.3344 | 0.7507 |

Figure 13: Recommended $\beta$ value for different user needs.

# I  PRICE OF MODELS

Table 9 shows the price we used for the self-served model. It was used by Together.ai at the time the experiment was conducted, and the price reflects the hardware requirement for the increasing model size.

Table 9: Dense and MoE large language model size and price per million tokens.

| Dense Models | | Mixture-of-Experts Models | |
|---|---|---|---|
| Model Size | Price | Model Size | Price |
| Up to 4B | $0.10 | Up to 56B | $0.60 |
| 4.1B – 8B | $0.20 | 56.1B – 176B | $1.20 |
| 8.1B – 21B | $0.30 | 176.1B – 480B | $2.40 |
| 21.1B – 41B | $0.80 | | |
| 41.1B – 80B | $0.90 | | |
| 80.1B – 110B | $1.80 | | |

