# OpenReview forum: "RouterArena: An Open Platform for Comprehensive Comparison of LLM Routers"
_ICLR.cc/2026/Conference — ICLR 2026 Poster_

### Official Review · Reviewer_vKx3 · 2025-10-31

**Soundness:** 1
**Presentation:** 3
**Contribution:** 3
**Rating:** 2
**Confidence:** 3

**Summary:**

This paper proposes a benchmark for LLM Routers that evaluates different routing systems along multiple axes—including *accuracy*, *cost*, *latency*, and *routing robustness*. Concretely, the authors curate *8,400* queries from 21 open-source datasets and, via an LLM-as-Judge procedure using DeepSeek-V3.1, assign each query to one of three difficulty levels: easy, medium, or hard. On the collected dataset, the paper conducts objective evaluations of both open-source routers (e.g., KNN- and MLP-based methods) and commercial routers (e.g., GPT-5, Azure Router).

**Strengths:**

1. It is necessary and practically valuable to objectively evaluate LLM routers, as this helps users who are unfamiliar with model details select configurations that best fit their needs.
2. The benchmark combines 5 evaluation perspectives (including cost, accuracy, robustness, latency, and routing optimality). In addition, it supports commercial routers, making the evaluation more complete than prior work.

**Weaknesses:**

### Weaknesses

1. Query selection bias. The query set consists primarily of objective questions and excludes creative/open-ended tasks. For objective questions, users often prioritize accuracy over cost; however, for open-ended or batch-processing tasks (e.g., large-scale data filtering), cost can be the primary concern, leading to different routing preferences. This preference shift can further affect the validity of conclusions drawn in the Experiments section. It is advisable to include open-ended queries and update the results and analyses accordingly.

2. Difficulty grading misalignment with real-world scenarios. Difficulty is determined solely via an LLM-as-Judge setup, which may introduce model-induced bias; human annotation should be incorporated. Moreover, the benchmark should expand to more realistic routing demands, such as *long-context generation*, *tool use*, *code agents*, and *deep research*. In real deployments, achieving good performance often involves *orchestrating multiple models* collaboratively rather than invoking a single model.

3. Limited comparability due to heterogeneous model pools. Different routers operate over different model pools. A router that can access a cheaper yet strong model may rank higher, but this does not necessarily demonstrate a better routing algorithm. Introducing a unified model pool and a corresponding leaderboard would improve comparability.

4. Evaluation dimensions remain limited. Additional aspects—such as *response time* or *response length*—should be considered. For example, if the pool contains a cheaper *reasoning* model and a more expensive *instruction* model, then under the cost formulation
$ \text{cost} = C_{\text{in}} \cdot N_{\text{in}} + C_{\text{out}} \cdot N_{\text{out}} $,
two routing strategies might achieve the *same cost* and *same accuracy*, yet differ meaningfully in behavior; how should their relative quality be judged?

### Notation

1. *All displayed formulas lack numbering*, which reduces ease of reference and clarity in discussion.

**Questions:**

See above weaknesses

---

> ### Author Response · Authors · 2025-11-24
> **Thank you! (1/4)**
>
> We thank the reviewer for the detailed review and recognition. Here, we address your feedback respectively.
>
> ---
>
> ## **W1. Bias toward objective queries ------ Beta handles preference; objective focus for reproducibility**
> > W1. Query selection bias. The query set consists primarily of objective questions and excludes creative/open-ended tasks. For objective questions, users often prioritize accuracy over cost; however, for open-ended or batch-processing tasks (e.g., large-scale data filtering), cost can be the primary concern, leading to different routing preferences. This preference shift can further affect the validity of conclusions drawn in the Experiments section. It is advisable to include open-ended queries and update the results and analyses accordingly.
>
>
> We thank the reviewer for the insightful thoughts. There are two distinct aspects here: (i) how users trade off cost vs. accuracy for their own preference, and (ii) how router behavior generalizes to qualitatively different tasks such as open-ended generation.
>
> For (i), RouterArena explicitly **models preference shifts via the β parameter** in our utility metric (Sec. 5.1). By varying β, we can move continuously from a cost-sensitive regime (large β) to an accuracy-dominant regime (small β). In other words, our leaderboard is not tied to a single “accuracy-first” preference; users who care more about cost (e.g., large-scale batch generation) can adopt a larger β, while users who prioritize accuracy can choose a smaller β. We clarify this point and provide more guidance on β selection in the revised version in Appendix H.
>
> For (ii), our current benchmark focuses primarily on closed-ended tasks for several reasons. (1) Objective questions have standardized answers, making the evaluation **more reliable and reproducible**. (2) Extending to open-ended tasks (e.g., story writing, brainstorming) is challenging because there is **no widely accepted, reliable automatic metric for such outputs**. Even LLM-as-a-judge can be unstable without well-designed rubrics and calibration, and **large-scale human evaluation is prohibitively expensive**. (3) **Prior router benchmarks** (e.g., RouterBench, RouterEval, FusionBench) predominantly use objective tasks just as we do.
>
>
> Even though **RouterArena makes the evaluation substantially more comprehensive than prior work**: we cover a wide range of domains and difficulty levels, **include non-QA tasks such as multi-lingual translation and reference reasoning**, and provide rich metrics together with a live leaderboard. We see robust open-ended routing evaluation as an important direction for **future work**, but believe that a carefully curated, objectively evaluable benchmark is a necessary first step. In this sense, we view RouterArena as laying the groundwork for future extensions that incorporate open-ended tasks once more reliable evaluation protocols become available.

---

> ### Author Response · Authors · 2025-11-24
> **Thank you! (2/4)**
>
> ## **W2. Difficulty grading & LLM-as-judge bias ------ Empirical difficulty and human validation**
>
> > W2. Difficulty grading misalignment with real-world scenarios. Difficulty is determined solely via an LLM-as-Judge setup, which may introduce model-induced bias; human annotation should be incorporated.
>
> We have revised Section 3 to reflect the role of the LLM-as-judge and our notion of "difficulty". **Bloom's Taxonomy in our benchmark is used only to characterize the cognitive skill of each query and to guide stratified sampling**; it is not used as the definition of difficulty in our analyses. **Difficulty is defined purely empirically**: for each query, we compute the number of models (out of 42) that answer it correctly and sort queries by this empirical difficulty. We then group queries into three bands—hard (≤4/42 models correct), medium (5–19/42), and easy (>19/42)—yielding 23.4%, 29.1%, and 47.5% of the dataset, respectively. As shown in Fig. 3–4, the overall and per-domain difficulty curves are smooth from the hardest to the easiest queries, indicating that the final 8.4k-query subset spans a broad and well-populated difficulty range for LLM routers.
>
> We also quantified the reliability of LLM as judges via **human annotation**. We conduct a human validation study on 5.4% of the data (450 queries), stratified by category and Bloom level. Eighteen volunteers (undergraduate and graduate students) label Bloom levels under the same protocol as the LLM. We observe 54.9% exact agreement and 76.7% agreement within ±1 Bloom level with the human majority vote, indicating that most disagreements are between adjacent levels in an inherently subjective taxonomy. Since Bloom labels only guide dataset construction, while all router evaluations rely on empirical accuracy across 42 LLMs, any model-induced bias from the LLM-as-judge has **a limited impact** on our reported difficulty grading or conclusions.
>
> ---
>
> ### **W2.2. More routing demands ------ Agents out-of-scope; coding included; LongBench-v2 added**
> > W2.2. Moreover, the benchmark should expand to more realistic routing demands, such as long-context generation, tool use, code agents, and deep research. In real deployments, achieving good performance often involves orchestrating multiple models collaboratively rather than invoking a single model.
>
>
> Thank you for the suggestion. We fully agree that tool use and multi-step "deep research" workflows are important aspects when using LLMs. However, **routers in such domains are still underexplored**. Today's routers primarily focus on mapping a user query to one backbone model in a model pool. Therefore, our design is tailored for the mainstream existing routers, but our framework **can be easily extended** when such routers become popular.
>
>
> **Coding agent is included** in our benchmark via queries from LiveCodeBench, which focus on program synthesis and code understanding. These tasks naturally differentiate routers that can reliably select strong coding models from those that cannot, providing a good backbone model in coding-agent settings.
>
> At the same time, we agree that **long-context generation** is a reasonable and important capability for LLM routers. To address this, we have **incorporated LongBench-v2** into our evaluation suite and report router performance in Table 1. We observe that GPT-5 achieves the highest accuracy on long-context tasks, while Azure-Router and NotDiamond offer slightly lower accuracy at substantially different cost points. Lightweight routers such as NIRT-BERT and MIRT-BERT also remain competitive in accuracy at much lower cost, whereas MLP and KNN attain reasonable accuracy only at very high cost, illustrating that naive over-use of large models is inefficient in the long-context regime. Overall, these results demonstrate that RouterArena can capture meaningful cost–accuracy trade-offs for long-context routing, complementing our core short-context evaluations.
>
> ---
>
> ### Table 1: RouterArena routers — long-context performance and cost.
>
> | Router       | Acc (%) | Cost / 1K Queries (USD) |
> |-------------|---------|--------------------------|
> | GPT-5       | 71      | 45.70                   |
> | NotDiamond  | 70      | 59.30                   |
> | Azure-Router| 67      | 9.54                    |
> | NIRT-BERT   | 60      | 6.60                    |
> | RouteLLM    | –       | –                       |
> | Graphrouter | 42      | 23.60                   |
> | vLLM-SR     | 42      | 64.80                   |
> | CARROT      | 25      | 4.10                    |
> | RouterDC    | 12      | 3.60                    |
> | MIRT-BERT   | 60      | 7.50                    |
> | MLP         | 65      | 286.70                  |
> | KNN         | 69      | 300.30                  |
>
> Note: RouteLLM couldn't process long context queries due to its text encoder only supports context lengths of 8192 tokens.

---

> ### Author Response · Authors · 2025-11-24
> **Thank you! (3/4)**
>
> ## **W3. Comparability of heterogeneous pools ------ Dynamic pools are more reasonable for the leaderboard; invariant metrics isolate algorithms**
> > W3. Limited comparability due to heterogeneous model pools. Different routers operate over different model pools. A router that can access a cheaper yet strong model may rank higher, but this does not necessarily demonstrate a better routing algorithm. Introducing a unified model pool and a corresponding leaderboard would improve comparability.
>
>
> Thank you for bringing up this insight. We believe evaluating routers as a whole is more reasonable for the leaderboard because:
>
> (1) **Model pool is critical for routers; the leaderboard should encourage designers to optimize their model selection.**
> When evaluating routers for a leaderboard, we should not restrict the model choice for routers, because the composition of the model pool is a key design decision in LLM routing. Many routers from our benchmark, such as MIRT-Bert and Azure-Router, derive their efficiency precisely from curating and leveraging lightweight cost-efficient models. As a leaderboard, we should encourage routers to discover and exploit new, cheaper, or more specialized models, something that static benchmarks fundamentally cannot support. A good analogy here is that when people evaluate LLMs on the LMArena leaderboard, people don't restrict the training data for models.
>
> (2) **RouterArena also provides model-pool–invariant metrics to evaluate algorithmic quality.**
> Although routers may differ in their model pools, RouterArena also supports evaluating the algorithmic quality of routers in a model-pool-invariant way. To achieve this, we designed two families of metrics, normalized deferral curves and optimality scores, that anchor each router to its own pool's theoretical upper performance bounds.
>
> **Normalized Deferral Curve (L. 358, Fig. 6).**
> To remove the influence of pool composition, we normalize each router's performance by the best-performing model *within its own pool*. This anchors all routers at a common reference point and then measures how efficiently the routing algorithm defers to cheaper models.
>
>
>
> **Optimality Scores (L. 370, Fig. 7).**
> To further isolate algorithmic behavior, we introduced three optimality metrics that measure how efficiently a router uses the resources it has access to:
>
> - **Optimal Selection Ratio:** fraction of queries where the router picks the *cheapest correct* model.
> - **Optimal Accuracy Ratio:** fraction of achieved accuracy over the theoretical optimum.
> - **Optimal Cost Ratio:** actual cost over the *minimal achievable cost* when answering correctly.
>
> With these metrics, we measured how efficiently a router operates over its model pool, which would provide insights into the algorithm design as opposed to the model pool construction.

---

> ### Author Response · Authors · 2025-11-24
> **Thank you! (4/4)**
>
> ## **W4. Additional metrics (Length/Time) ------ Length added; Time is unreliable**
> > W4. Evaluation dimensions remain limited. Additional aspects—such as response time or response length—should be considered. For example, if the pool contains a cheaper reasoning model and a more expensive instruction model, then under the cost formulation cost = C_in * N_in + C_out * N_out, two routing strategies might achieve the same cost and same accuracy, yet differ meaningfully in behavior; how should their relative quality be judged?
>
>
> Thank you for the suggestion. We agree that response length and response time are important behavioral characteristics. Following your comment, we **added average response length (in output tokens)** to the RouterArena leaderboard and show the updated table at the end. This metric reflects how frequently a router selects more reasoning-heavy models, e.g., GPT-5 and Not Diamond often yield longer outputs. We do not treat longer answers as better; the metric simply provides another axis for users to choose between concise and detailed responses.
>
> By contrast, although we initially considered wall-clock response time, we ultimately decided not to include it because it is extremely hard to measure fairly and reproducibly. RouterArena evaluates heterogeneous routers that depend on proprietary APIs (e.g., Claude Sonnet-4, GPT-5) running on different hardware over the public internet. In this setting, observed latency is dominated by provider infrastructure and network conditions, not the router logic itself. We therefore focus on router-intrinsic latency, the overhead introduced before issuing the LLM call, which we report as our “router latency” metric.
>
> Regarding your concrete example where two routers that achieve the same cost and accuracy but differ in output behavior, our view is that such routers are **Pareto-equivalent on the primary axes**. In practice, users adopt routers either (i) to reduce inference cost by using cheaper models, or (ii) to improve accuracy by selecting domain-specialized models. RouterArena provides tunable $\beta$ to let users adjust the cost-accuracy trade-off.
>
> ---
>
> ### Table 2: RouterArena routers — average generation length and cost
>
> | Router       | Avg Gen Length | Cost/M Tokens |
> |-------------|----------------|---------------|
> | GPT-5       | 970.65         | $10.00        |
> | NotDiamond  | 390.05         | $9.18         |
> | Azure-Router| 270.90         | $1.79         |
> | NIRT-BERT   | 215.82         | $1.33         |
> | RouteLLM    | 140.24         | $0.68         |
> | Graphrouter | 132.24         | $0.81         |
> | vLLM-SR     | 109.27         | $9.03         |
> | CARROT      | 105.56         | $12.58        |
> | RouterDC    | 102.83         | $0.20         |
> | MIRT-BERT   | 89.16          | $1.00         |
> | MLP         | 81.06          | $29.75        |
> | KNN         | 79.80          | $26.31        |
>
> ---
>
> ## **Notation**
> > All displayed formulas lack numbering, which reduces ease of reference and clarity in discussion.
>
> Thanks for the suggestions! We have added the number labels for each equation in the paper.

---

### Official Review · Reviewer_ZQTb · 2025-10-31

**Soundness:** 3
**Presentation:** 4
**Contribution:** 2
**Rating:** 2
**Confidence:** 3

**Summary:**

This paper introduces RouterArena, which the authors claim to be the first comprehensive open platform for evaluating and comparing large language model (LLM) routers. The platform aims to meet the growing need for systematic router evaluation and provides: (1) a principled dataset consisting of 8,400 queries spanning 9 domains and 44 categories, organized using the Dewey Decimal Classification system and incorporating Bloom’s taxonomy to define difficulty levels; (2) a comprehensive set of evaluation metrics covering accuracy, cost, routing optimality, robustness, and latency; and (3) an automated evaluation framework that supports both open-source and commercial routers. The authors evaluate 12 representative routers and present a multidimensional leaderboard summarizing their overall performance.

**Strengths:**

S1. The paper addresses a critical gap in the LLM ecosystem, the research motivation is clear and timely.
S2. The evaluation metrics are comprehensive, with five well-defined dimensions that reflect real-world deployment considerations.
S3. The automated framework enables dynamic leaderboard updates and supports the evaluation of both open-source and commercial routers.

**Weaknesses:**

W1. The overall contribution of the paper is limited.
W2. For commercial routers, internal routing decisions are inaccessible, making many metrics uncomputable and thus limiting a full evaluation.
W3. The use of DeepSeek-V3.1 for automated difficulty annotation may introduce systematic bias; no quantitative bias analysis or annotation-consistency study is provided.
W4. The robustness test is limited, and the actual evaluation method appears inconsistent with the definition of robustness given in Section 4.
W5. The paper spends substantial space describing the construction of a dataset with distinct difficulty levels but does not subsequently analyze results based on these levels.

**Questions:**

Q1. Regarding the validation of Bloom’s taxonomy classification—was any human sampling verification conducted, and how consistent were the results with the LLM’s judgments?
Q2. The robustness definition in Section 4 differs from the implementation described in Section 6.2 (“adding irrelevant keywords”). If only keyword-insertion was used, does this cause inconsistency between the definition and the actual test?
Q3. Figure 6 looks more like a scatter plot than a curve plot. Would the authors consider renaming it for accuracy?
Q4. The paper builds a dataset with three difficulty levels, but all evaluations are aggregated. Why not provide router performance broken down by difficulty level?
Q5. For commercial routers that cannot expose routing decisions, is there a plan to design alternative indicators to enhance ranking reliability?

---

> ### Author Response · Authors · 2025-11-24
> **Thanks! (1/3)**
>
> We thank the reviewer for your careful review and recognition. We recognized the main concern of your review and provided additional results and discussions. Here, we address your concerns and questions in detail.
>
> ---
>
> ## **W1. Contribution ------ This is the first unified and principally constructed router leaderboard**
> > W1. The overall contribution of the paper is limited.
>
> We would like to clarify our key contributions in the following:
> 1. **Addressing the limitations of existing router benchmarks.** As shown in L.120 Table 1, current benchmarks lack comprehensive coverage of query categories, do not distinguish difficulty levels, rely on overly simple evaluation metrics, and do not evaluate commercial routers. Moreover, they do not offer a clear ranking that people can rely on when selecting routers.
> 2. **Providing the first unified and comprehensive evaluation framework for LLM routers.** Routers are emerging rapidly, and even after our submission, several new routers [1, 2] have appeared. Yet there remains no standardized way to compare them. Researchers still lack a comprehensive and unified methodology for evaluating routers across both academic and commercial systems. RouterArena is the first framework to fill this gap.
> 3. **Establishing a principled foundation for future benchmark construction.** RouterArena introduces an extensible category structure that makes it straightforward to incorporate additional data and to assign difficulty levels in a systematic way. This design offers a clear guideline for constructing future router benchmark datasets and supports continued expansion as the field evolves.
>
> We hope this clarification addresses your concern about the contribution; if there are specific aspects you still find limited, we would greatly appreciate further suggestions.
>
> [1] RADAR: REASONING–ABILITY AND DIFFICULTY AWARE ROUTING FOR REASONING LLMS, arXiv:2509.25426
>
> [2] Adaptive LLM Routing Under Budget Constraints, arXiv:2508.21141
>
> ---
>
>
> ## **W2/Q5. Black-box/commercial router evaluation ------ Primary metrics are obtainable, and additional metrics are possible.**
> > W2. For commercial routers, internal routing decisions are inaccessible, making many metrics uncomputable and thus limiting a full evaluation.
> > Q5. For commercial routers that cannot expose routing decisions, is there a plan to design alternative indicators to enhance ranking reliability?
>
> Thank you for this question. Among the three commercial routers we evaluate, **only GPT-5 behaves as a true black box**: it returns final answers and billing information, but not the internal model IDs. For NotDiamond and Azure-Router, we are able to log their model selections, so we compute all metrics that require routing decisions (**optimality and robustness scores updated in Fig. 8 & 9** respectively). For the latency analysis, we exclude commercial routers (GPT-5, Azure-Router, and NotDiamond) because they are accessed via remote APIs, whereas academic routers are hosted locally; including them would conflate model latency with additional, highly variable network overhead and lead to an unfair comparison.
>
> For router evaluation, the most important score is the **Arena score, which is defined in terms of accuracy and monetary cost** (Sec. 5.1), and this is observable for **every router**. So the main ranking axis is unaffected by whether internal routing decisions are visible. The additional behavior-focused metrics (e.g., optimality and robustness) are mainly diagnostic, and we view **the inclusion of commercial black-box routers as a strength rather than a limitation**. RouterArena is, to our knowledge, the first benchmark to evaluate such systems side-by-side with academic routers, thereby filling a key gap in prior work where commercial routers were omitted entirely.

---

> ### Author Response · Authors · 2025-11-24
> **Thanks! (2/3)**
>
> ## **W3/Q1. LLM-as-judge bias ------ Empirical difficulty evaluation and human validation provided**
> >W3. The use of DeepSeek-V3.1 for automated difficulty annotation may introduce systematic bias; no quantitative bias analysis or annotation-consistency study is provided.
> >Q1. Regarding the validation of Bloom's taxonomy classification—was any human sampling verification conducted, and how consistent were the results with the LLM's judgments?
>
> Thank you for the thoughtful comment. We have revised Section 3 to clarify the role of Bloom's Taxonomy and the LLM-as-judge labels in our benchmark.
>
> Bloom's Taxonomy measures the cognitive skill required by a question (from remembering to evaluating), and in RouterArena we **use it only as a coarse before ensuring coverage of diverse cognitive skills**. The **actual difficulty used in all analyses is defined empirically**: for each query, we compute the average accuracy over 42 LLMs (from a range of small to large models, details in Appendix C.4) and sort queries by this empirical difficulty.
>
> Concretely, we define **three difficulty bands based on the number of models that answer a query correctly**: hard (≤4/42 LLMs), medium (5~19/42), and easy (>19/42), resulting in 23.4%, 29.1%, and 47.5% of the dataset, respectively (Fig. 3 in our paper). As shown in our revised Fig. 4, the overall difficulty curve (sorted by empirical accuracy) is smooth from hardest to easiest queries, and each Dewey domain exhibits a similar smooth trend without large gaps. This indicates that the final 8.4k-query subset we used spans a broad and well-populated difficulty range for LLM routers.
>
> We also conducted a **human validation study** by sampling roughly 5% of the data stratified over category and Bloom level (450 questions). Eighteen volunteers (undergraduate and graduate students) labeled Bloom levels under the same protocol as the judging LLM. We observe 54.9% exact agreement and 76.7% agreement within ±1 Bloom level with the human majority vote, reflecting that disagreements are mostly between adjacent levels in an inherently subjective taxonomy. **Importantly**, we do not use Bloom labels as ground-truth difficulty in any metric; they only guide dataset construction. For router evaluation, we need queries that distinguish models in the zoo, and our empirically derived difficulty levels are designed to do exactly that.
>
> ---
>
> ## **W4/Q2. Robustness implementation mismatch ------ We aligned the definition with the new experiment**
> >W4. The robustness test is limited, and the actual evaluation method appears inconsistent with the definition of robustness given in Section 4.
>
> >Q2. The robustness definition in Section 4 differs from the implementation described in Section 6.2 (“adding irrelevant keywords”). If only keyword-insertion was used, does this cause inconsistency between the definition and the actual test?
>
>
>
> Thank you for the careful reading. **We have now aligned the evaluation procedure with our robustness definition.** Specifically, we assign equal weight to the four noisy variants described in the definition. For each category and its subcategories, we randomly sampled 5% of the entire dataset, which is 420 examples, and generated perturbed queries using GPT-4o. The prompt we used is provided in Appendix G. After regenerating all noisy queries, we reran the robustness evaluation and found that the open-source routers that rely on BERT for latent representations (e.g., vLLM-SR, MIRT-BERT, and NIRT-BERT) exhibit noticeably weaker robustness than the others. This aligns with prior findings that BERT is sensitive to surface-level perturbations.[1]
>
> ---
>
> ### Table 1: Robustness score on rotuers.
> | Router      | Robustness |
> | ----------- | ---------- |
> | vLLM-SR | 35.00      |
> | NIRT-BERT        | 49.29      |
> | Auzre-Router       | 54.07      |
> | NotDiamond  | 55.91      |
> | MIRT-BERT        | 61.19      |
> | MLP         | 80.00      |
> | KNN         | 83.33      |
> | Kmeans      | 84.52      |
> | RouterDC    | 85.24      |
> | CARROT      | 89.05      |
> | GraphRouter | 94.29      |
> | RouteLLM    | 100.00     |
>
> [1] Is BERT Really Robust? A Strong Baseline for Natural Language Attack on Text Classification and Entailment. arXiv:1907.11932

---

> ### Author Response · Authors · 2025-11-24
> **Thanks! (3/3)**
>
> ## **W5/Q4. Analysis by difficulty ------ Comprehensive breakdown added**
> > W5. The paper spends substantial space describing the construction of a dataset with distinct difficulty levels but does not subsequently analyze results based on these levels.
> > Q4. The paper builds a dataset with three difficulty levels, but all evaluations are aggregated. Why not provide router performance broken down by difficulty level?
>
> Thank you for pointing this out. In the revision, we now **provide a comprehensive breakdown of router performance by difficulty level** (Easy / Medium / Hard), including both accuracy and average cost per 1K queries (Table 6 in the paper and we paste it below).
>
> This analysis confirms that our empirical difficulty labels behave as intended: all routers achieve very high accuracy on Easy queries but suffer substantial degradation on Hard queries. On Easy examples, lightweight routers such as MIRT-BERT, CARROT, and vLLM-SR nearly match GPT-5 while being much cheaper, whereas on hard queries, GPT-5 remains the strongest, with Azure-Router and NotDiamond offering the next-best cost–accuracy trade-offs. We also observe that some routers (e.g., GPT-5, Azure-Router, MIRT-BERT, NotDiamond) allocate significantly more budget to Hard than Easy queries, while others show almost flat cost across difficulty, suggesting room for more difficulty-aware routing strategies.
>
> ---
>
> ### Table 2: Router performance by difficulty level (accuracy and cost per 1k queries).
>
> | Router        | Overall             | Easy                | Medium              | Hard                 |
> |--------------|---------------------|---------------------|---------------------|----------------------|
> | GPT-5        | 74.0% ($14.02)      | 95.1% ($5.68)       | 68.6% ($14.80)      | 27.5% ($35.73)       |
> | Azure-Router | 68.1% ($0.54)       | 93.3% ($0.30)       | 59.5% ($0.63)       | 17.9% ($1.05)        |
> | NotDiamond   | 68.0% ($9.34)       | 92.5% ($6.34)       | 61.9% ($10.29)      | 15.1% ($15.16)       |
> | vLLM-SR      | 67.3% ($1.67)       | 95.3% ($1.56)       | 57.9% ($1.70)       | 8.7% ($1.87)         |
> | CARROT       | 67.2% ($2.06)       | 95.0% ($1.79)       | 58.0% ($2.26)       | 9.0% ($2.36)         |
> | MIRT-BERT    | 66.9% ($0.15)       | 95.9% ($0.10)       | 56.8% ($0.17)       | 7.1% ($0.26)         |
> | NIRT-BERT    | 62.0% ($0.69)       | 93.7% ($0.53)       | 47.0% ($0.78)       | 4.7% ($0.90)         |        |
> | MLP          | 61.6% ($4.83)       | 93.2% ($3.54)       | 46.6% ($5.74)       | 4.6% ($6.52)         |
> | KNN          | 58.7% ($4.27)       | 89.2% ($3.08)       | 43.6% ($5.13)       | 4.5% ($5.77)         |
> | Graphrouter  | 57.0% ($0.34)       | 89.7% ($0.25)       | 39.3% ($0.39)       | 2.4% ($0.46)
> | RouteLLM     | 47.0% ($0.27)       | 79.8% ($0.20)       | 24.4% ($0.33)       | 2.0% ($0.32)         |
> | RouterDC     | 32.0% ($0.07)       | 54.9% ($0.06)       | 15.3% ($0.08)       | 2.2% ($0.09)         |
>
> ---
>
> ## **Q3. Figure naming ------ Renamed to Normalized Scatter Plot**
> > Q3. Figure 6 looks more like a scatter plot than a curve plot. Would the authors consider renaming it for accuracy?
>
> Thanks for the suggestion! We have changed its name to "Normalized Scatter Plot".

---

### Official Review · Reviewer_LQ8R · 2025-11-01

**Soundness:** 2
**Presentation:** 3
**Contribution:** 3
**Rating:** 8
**Confidence:** 3

**Summary:**

This paper introduces "RouterArena," a new open-source platform for benchmarking Large Language Model (LLM) routers. The authors argue that as the LLM ecosystem has produced numerous models with varying costs and capabilities, routers have become essential for selecting the right model for a given query. This, in turn, has created a new need for a standardized way to evaluate the routers themselves.


An automated framework and leaderboard designed to be "live," allowing researchers to submit new routers (both open-source and commercial) for evaluation and comparison.

**Strengths:**

1. The paper addresses a practical problem. As model routing becomes a standard component in AI stacks, the need for a robust, standardized benchmark to compare routers is high. This work is well-motivated and highly relevant to the community.

2. Principled Dataset Construction: A major strength of this paper is its novel and well-justified dataset construction.

3. Novelty: An automated, "live" platform and leaderboard, distinct from a static dataset. It is designed for continuous benchmarking and community engagement, allowing researchers to submit and compare new open-source and commercial routers.

4. Comprehensive, Multi-Dimensional Evaluation: The paper correctly identifies that router performance is multi-faceted. The inclusion of Routing Optimality is a key metric, as it reframes the goal from just "being correct" to "being correct efficiently." Measuring Robustness and Latency further strengthens the benchmark's utility for real-world applications.

5. Interesting Initial Analysis and Insights: The initial evaluation of 12 routers provides valuable insights. The findings—that commercial routers do not necessarily lead, that all routers are inefficient, and that performance is a complex trade-off—are important takeaways for the field.

**Weaknesses:**

Discussion of Benchmarking Philosophy: The paper positions itself as superior to prior work like RouterBench (Table 1), but it misses an opportunity for a deeper discussion. RouterArena evaluates live routers (a "hot" evaluation), whereas RouterBench uses a large, static dataset of pre-computed outcomes for "offline" evaluation. This offline approach is significantly cheaper and faster for iterating on router designs, presenting a different benchmarking philosophy. A more nuanced discussion of the pros and cons of these different approaches would improve the paper's contribution.

Dataset Scale: While the dataset is principled in its design, its size (~8,400 queries) is a potential limitation. When spread across 44 categories and 3 difficulty levels, some cross-sections may be too small to draw statistically significant conclusions. The paper would be stronger if it addressed this limitation or included an analysis of the sample size per category.

**Questions:**

See weaknesses.

---

> ### Author Response · Authors · 2025-11-24
> **Thanks!**
>
> Thank you for providing positive reviews of our paper. We hope to address your concerns below!
>
> ---
>
>
> ## **W1. Static vs. Dynamic pools / Online vs. Offline evaluation.**
> > W1. Discussion of Benchmarking Philosophy: The paper positions itself as superior to prior work like RouterBench (Table 1), but it misses an opportunity for a deeper discussion. RouterArena evaluates live routers (a "hot" evaluation), whereas RouterBench uses a large, static dataset of pre-computed outcomes for "offline" evaluation. This offline approach is significantly cheaper and faster for iterating on router designs, presenting a different benchmarking philosophy. A more nuanced discussion of the pros and cons of these different approaches would improve the paper's contribution.
>
> We appreciate the insightful thoughts. RouterBench follows an offline design, where a large, static dataset of pre-computed model responses is released once and then reused. This style is valuable for researchers iterating on routing algorithms: it is cheap to run, fully reproducible, and isolates algorithmic effects under a fixed model pool and fixed query distribution.
>
> By contrast, RouterArena is explicitly designed as a live leaderboard and evaluation. Our goal is to let users compare deployed routers as systems with the latest routers in the world. This has three main implications:
> 1. **Inclusion of commercial/closed-source routers.** Many routers we benchmark (e.g., GPT-5, Azure Model Router, NotDiamond) only expose an API. For these systems, pre-computed model traces are not available, so an offline–log benchmark cannot evaluate them directly. Our online setup lets us benchmark these services under the same protocol as academic routers.
> 2. **Supporting evolving, specialized model pools.** The LLM ecosystem is rapidly evolving. A static offline benchmark necessarily freezes the model pool, so it cannot reflect future models or reward routers that successfully exploit them. In contrast, a live leaderboard explicitly encourages router designers to exploit these models in their pools, making routing decisions more meaningful in practice.
> 3. **Measuring system-level metrics that require live calls.** RouterArena reports not only accuracy and cost, but also router overhead/latency, robustness under perturbed prompts, and cost computed using current provider prices. These quantities fundamentally depend on live execution rather than a fixed set of cached responses.
>
> We agree that these two philosophies are complementary rather than competing: offline corpora like RouterBench are ideal for cheap, controlled algorithm development, while RouterArena is aimed at a live, system-level leaderboard that tracks end-to-end behavior of both open-source and commercial routers. We revised the "Related Work" section to reflect these design philosophy differences.
>
> ---
>
> ## **W2. Dataset size limitations ----- An efficient subset with sufficient coverage can be easily extended.**
> > W2. Dataset Scale: While the dataset is principled in its design, its size (~8,400 queries) is a potential limitation. When spread across 44 categories and 3 difficulty levels, some cross-sections may be too small to draw statistically significant conclusions. The paper would be stronger if it addressed this limitation or included an analysis of the sample size per category.
>
>
> Thank you for bringing up this point. The current dataset of 8,400 examples is **a curated subset of the original collection of more than 30,000 queries**. Because router evaluation requires running LLM inference for every single example, we reduced the dataset for efficiency. Even after this reduction, the dataset still **clearly differentiates both performance and cost across routers**, largely due to its broad coverage across domains and difficulty levels.
>
> In addition, in L. 177 Fig. 3 (left panel), the inner ring includes numeric labels indicating the number of samples in each category. Except for a few niche categories (such as Animals, Management and Public Relations, and Social Problems) where the available datasets are inherently limited and thus result in smaller sample sizes, **all other categories contain sufficient data and are well represented**.
>
> Moreover, we believe RouterArena provides a principled blueprint for building future router evaluation datasets. Our category design is modular and extensible, making it easy to incorporate additional data or new domains as needed.

---

### Official Review · Reviewer_T3An · 2025-11-02

**Soundness:** 3
**Presentation:** 3
**Contribution:** 3
**Rating:** 6
**Confidence:** 3

**Summary:**

The paper tackles a timely problem—standardized router evaluation—with a clear systemization (dataset + metrics + framework) and substantive empirical coverage (open-source and commercial).

**Strengths:**

S1. Well-scoped problem & gap analysis.

S2. Good dataset, metric, and evaluation design.

S3. Comprehensive experiments and visualizations.

**Weaknesses:**

W1. The $\log 2$ cost normalization with fixed ( $c_{min}⁡ = 0.0044$, $c_{max} ⁡ = 200$ ) and $\beta=0.1$ may bias rankings toward certain price bands; no sensitivity analysis shown in Sec. 5.

W2. LLM-as-judge labeling (DeepSeek-V3.1) lacks human validation studies or inter-rater checks.

**Questions:**

I think this is a very interesting topic. Specifically, I have the following two questions:

Q1. Your evaluation reports per-query accuracy as a binary outcome and aggregates it—without explicit weighting by Bloom difficulty levels—into the composite Arena score $S_{i,\beta}$.
-  Is “completion” strictly 0 or 1 correctness per query? If so, why not support rubric-based partial credit for partially solved answers (e.g., correct plan but incomplete final step)?
- Some longer (higher-token) first-round answers can enable success in later turns. Do you plan a multi-turn setting that measures cross-round utility versus added first-round cost (e.g., a “round-2 success gain” vs. round-1 token spend)?

Q2. Among answers that are equally correct, a longer response may offer clearer reasoning, citations, or actionable steps, potentially increasing user satisfaction even at a higher token cost. Do you collect any user-satisfaction / explanation-quality signal (human Likert ratings or a calibrated LLM-as-judge) in addition to accuracy/cost?

---

> ### Author Response · Authors · 2025-11-24
> **Thank you! (1/2)**
>
> We thank the reviewer for your detailed review and recognition. Here, we address your concerns and questions.
>
> ---
>
>
> ## **W1. Cost normalization bias & Beta sensitivity ------ We provide dynamic normalization & beta tuning**
>
> > W1. The log2 cost normalization with fixed (c_min, c_max) and beta = 0.1 may bias rankings toward certain price bands; no sensitivity analysis shown in Sec. 5.
>
> In our framework, $c_{max}$ and $c_{min}$ are not intended to be static global constants; instead, they are **tunable for flexibility**. To prevent the most expensive router from being normalized to zero, for now, we set $c_{max}$ and $c_{min}$ to the prices of the most expensive and cheapest models on the market. However, whenever a model's actual cost really falls below $c_{min}$ or exceeds $c_{max}$, the only thing we need to do is to adjust the $c_{min}$ and $c_{max}$, and re-calculate the ArenaScore so that all normalized costs remain within $(0,1)$. Note that this **doesn't require re-running the experiment**.
>
> The parameter $\beta$ is **also adjustable** to modulate how much weight is assigned to cost compared to accuracy. Specifically, we made the $\beta$ tunable on our leaderboard for users (will have a website open to users). Here we give three example settings:
>
> * Example 1: $\beta$ = 0.01, the ratio of Accuracy to Cost is 100, accuracy-dominated ranking for users who do not consider cost to be a key constraint.
> * Example 2: $\beta$ = 0.1, the ratio of Accuracy to Cost is 10, trade-off balanced ranking for users who primarily care about accuracy but still operate under budget constraints.
> * Example 3: $\beta$ = 1, the ratio of Accuracy to Cost is 1, equally weighted ranking for users who have a limited budget.
>
> ---
>
> ### Table 1: Leaderboard by different values of Beta.
> |Router|Rank & ArenaScore @ β=0.01|Rank & ArenaScore @ β=0.1|Rank & ArenaScore @ β=1|Accuracy|Norm.Cost|
> |-----------------|------------------------------|-----------------------------|---------------------------|--------|---------|
> |GPT-5|1 (0.7282)|5 (0.6272)|12 (0.3688)|0.7428|0.2453|
> |Azure-Router|2 (0.6780)|2 (0.6640)|2 (0.6010)|0.6798|0.5386|
> |MIRT-BERT|3 (0.6731)|1 (0.6729)|1 (0.6718)|0.6731|0.6705|
> |CARROT|4 (0.6682)|3 (0.6386)|6 (0.5219)|0.6720|0.4266|
> |vLLM-SR|5 (0.6633)|4 (0.6385)|5 (0.5368)|0.6665|0.4494|
> |NotDiamond|6 (0.6565)|8 (0.5935)|11 (0.3998)|0.6651|0.2858|
> |RouteLLM|7 (0.6175)|9 (0.5800)|10 (0.4440)|0.6224|0.3451|
> |NIRT-BERT|8 (0.6151)|6 (0.6083)|4 (0.5764)|0.6159|0.5416|
> |MLP|9 (0.6143)|10 (0.5780)|9 (0.4449)|0.6191|0.3472|
> |GraphRouter|10 (0.6070)|7 (0.6054)|3 (0.5976)|0.6072|0.5884|
> |KNN|11 (0.5867)|11 (0.5577)|8 (0.4464)|0.5905|0.3588|
> |RouterDC|12 (0.3362)|12 (0.3522)|7 (0.4627)|0.3344|0.7507|
>
> Thanks for your suggestion again! We have revised the paper to incorporate the changes in Appendix H.
>
> ---
>
> ## **W2. Lack of human validation ------ LLM-as-a-judge only for guidance; difficulty is defined empirically; human study provided.**
>
> >W2. LLM-as-judge labelling (DeepSeek-V3.1) lacks human validation studies or inter-rater checks.
>
> We agree that the role of LLM-as-judge should be clearly specified, and we have revised Section 3 accordingly. Bloom's Taxonomy in our benchmark is **used only to characterise the cognitive skill** of each query and to guide stratified sampling; it is **not used as the definition of difficulty** in our analyses. **Difficulty is defined empirically**: for each query, we compute the number of models (out of 42 models with various sizes and capabilities used by existing routers, details in Appendix C.4) that answer it correctly and sort queries by this empirical difficulty. We then group them into three bands—hard (≤4/42 models correct), medium (5–19/42), and easy (>19/42)—yielding 23.4%, 29.1%, and 47.5% of the dataset, respectively, with smooth difficulty curves overall and per domain (Fig. 3–4).
>
> We also quantified the reliability of LLMs as judges via a **human validation study** on 5.4% of the data (450 queries), stratified by category and Bloom level. Eighteen volunteers (undergraduate and graduate students) labelled Bloom levels under the same protocol as the LLM. We observe 54.9% exact agreement and 76.7% agreement within ±1 Bloom level with the human majority vote, indicating that most disagreements are between adjacent levels in an inherently subjective taxonomy. Since Bloom labels only guide dataset construction, while all router evaluations rely on empirical accuracy across 42 LLMs, any bias from the LLM-as-judge has **limited impact** on our reported results.

---

> ### Author Response · Authors · 2025-11-24
> **Thank you! (2/2)**
>
> ## **Q1: Per-query accuracy and difficulty levels**
> ### **Q1.1 Rubric-based partial credit ------ Already included**
> >Q1.1 Is “completion” strictly 0 or 1 correctness per query? If so, why not support rubric-based partial credit for partially solved answers (e.g., correct plan but incomplete final step)?
>
>
> Thank you for pointing out the partial credit. **Not all of the queries in our benchmark are binarily scored**. In fact, around 10% of the queries used rule-based metrics, such as Meteor score, which is used for comparing the semantic similarity between two pieces of text. For the remaining datasets, we intentionally follow the original benchmark scoring schemes (typically 0/1 instance-level correctness) and avoid introducing new rubrics that would lack the empirical validation and reliability analyses already provided by the underlying benchmarks.
>
> RouterArena's design, however, makes it easy to plug in new tasks and metrics: as long as a benchmark provides an automatic evaluation procedure for LLM outputs, it can be directly integrated into our router evaluation pipeline.
>
> ### **Q1.2 Multi-turn utility ------ Future work; single-turn foundation first**
> > Q1.2 Some longer (higher-token) first-round answers can enable success in later turns. Do you plan a multi-turn setting that measures cross-round utility versus added first-round cost (e.g., a “round-2 success gain” vs. round-1 token spend)?
>
> Thank you for bringing up this insight. We agree that multi-turn interaction introduces worth-noticing dimensions. However, **we intentionally scope this work to single-turn routing** and leave the multi-turn LLM routing to future work for two reasons.
>
> First, **nearly all existing LLM routers are designed and evaluated in a single-turn setting**, where they encode the input query into an embedding and perform routing based on this single query embedding. Prior benchmarks such as RouterBench [1], EmbedLLM [2] (ICLR'25), FusionBench [3], and RouterEval [4] (EMNLP'25) all assume single-turn queries. Moreover, given that almost every router is still designed for single-turn conversation, we focus on the single-turn routing evaluation framework first in this work.
>
> Second, **a reliable multi-turn benchmark requires a strong single-turn foundation**. We envision RouterArena as the first step to a comprehensive single-turn router benchmark where we improved over prior works at domain coverage, difficulty separation, metric completeness, and support for commercial routers. A multi-turn routing benchmark would be a great follow-up work in the future.
>
> [1] RouterBench: A Benchmark for Multi-LLM Routing System. arXiv:2403.12031.
>
> [2] EmbedLLM: Learning Compact Representations of Large Language Models. arXiv:2410.02223.
>
> [3] FusionBench: A Comprehensive Benchmark of Deep Model Fusion. arXiv:2406.03280.
>
> [4] RouterEval: A Comprehensive Benchmark for Routing LLMs to Explore Model-level Scaling Up in LLMs. arXiv:2503.10657
>
> ---
>
> ## **Q2. User-experience-related signals ------ Objective and reproducible metrics are the primary focus; additional signals added**
> > Q2. Among equally correct answers, a longer response may offer clearer reasoning, citations, or actionable steps, potentially increasing user satisfaction even at a higher token cost. Do you collect any user-satisfaction / explanation-quality signal (human Likert ratings or a calibrated LLM-as-judge) in addition to accuracy/cost?
>
> We appreciate the insightful thoughts. **RouterArena currently focuses on reproducible metrics** (accuracy, cost, and a few behavioral metrics such as routing optimality, robustness, and latency).
>
> We didn't collect user-satisfaction / explanation-quality signals because **user preferences are inherently subjective and application-specific**, rather than universal. Different users could have very different preferences: some favor short, to-the-point answers, while others explicitly want long, step-by-step, "chain-of-thought" style responses. A single, benchmark-wide "user satisfaction" or "explanation quality" score would implicitly hard-code one particular notion of a "good" answer into RouterArena, instead of allowing practitioners to select routers that best match their own preference profiles. Moreover, obtaining a robust satisfaction estimate would **require large-scale, continuously refreshed human feedback** (as in Chatbot-Arena-style platforms), which is closer to a **commercial product/operations effort than to academic research**.
>
> We additionally report the **average generation length** as a dimension that surfaces the trade-off highlighted by the reviewer: among routers with similar accuracy and cost, some tend to produce shorter, more concise answers, whereas others systematically generate longer, more detailed responses. This gives users extra information when comparing routers, without forcing a single global preference into the core benchmark.

---

### Author Response · Authors · 2025-11-29
**Summary for AC**

Dear AC,

Below, we present a summary of our contributions along with our responses to the reviewers' concerns.


---

# **Our contribution**
We introduce RouterArena, the first open platform (leaderboard) to evaluate LLM routers systematically, which provides:
1) A principly-constructed dataset for routing evaluation with diverse domain coverage,
2) Comprehensive metrics capturing various aspects of routing behavior,
3) An automatic evaluation pipeline for both academic and commercial routers, and
4) A live leaderboard enabling ongoing, reproducible comparison.

---

# **Reviewers' recognition**

All reviewers recognized the importance of the problem and the value of our contributions.

- **Comprehensive LLM router evaluation is well-motivated and important**:
    - Reviewer `T3An` (6): *"The paper tackles a timely problem—standardized router evaluation."*
    - Reviewer `LQ8R` (8): *"The need for a robust, standardized benchmark to compare routers is high."*
    - Reviewer `ZQTb` (2): *"The paper addresses a critical gap, the research motivation is clear and timely."*
    - Reviewer `vKx3` (2): *"It is necessary and practically valuable to objectively evaluate LLM routers."*
- **Our dataset construction is principled and well-justified**:
    - Reviewer `T3An` (6): *"Good dataset, metric, and evaluation design."*
    - Reviewer `LQ8R` (8): *"[...] novel and well-justified dataset construction."*
    - Reviewer `ZQTb` (2): *"A principled dataset consisting of 8400 queries [...]"*
- **Our metrics are realistic and novel**:
    - Reviewer `T3An` (6): *"A clear systemization (dataset + metrics + framework)."*
    - Reviewer `LQ8R` (8): *"Correctly identifies that router performance is multi-faceted. The inclusion of Routing Optimality is a key metric. [...] Measuring Robustness and Latency further strengthens the benchmark's utility for real-world applications. "*
    - Reviewer `ZQTb` (2): *"The evaluation metrics are comprehensive, [...] reflect real-world deployment considerations."*
- **Our eval pipeline and leaderboard are comprehensive, necessary for the community, and more complete than prior works**:
    - Reviewer `T3An` (6): *"Comprehensive experiments and visualizations."*
    - Reviewer `LQ8R` (8): *"An automated, 'live' platform and leaderboard, distinct from a static dataset, designed for continuous benchmarking and community engagement."*
    - Reviewer `ZQTb` (2): *"The automated framework enables dynamic leaderboard updates and supports the evaluation of both open-source and commercial routers."*
    - Reviewer `vKx3` (2): *"It supports commercial routers, making the evaluation more complete than prior work."*

---

# Responses to reviewers' concerns

We have addressed **all** weaknesses/questions for each reviewer by clarifying misunderstandings and **adding six new experiment results**.

1) **Role of LLM-as-a-judge and difficulty levels (Provided human study, new empirical results, and new breakdown results)**: We clarify that LLM-as-a-judge is only used to recognize diverse cognitive skills (Bloom's Taxonomy) needed for each query. In fact, difficulty labels are empirically determined by the overall correctness of 42 LLMs. We provided a human study for the LLM-as-a-judge and added the empirical difficulty results. We also offered a new set of results with a comprehensive breakdown of router performance by the above-defined difficulty level.
    > Reviewer `T3An (6): W2`, Reviewer `ZQTb (2): W3/Q1, W5/Q4`, Reviewer `vKx3 (2): W2`
2) **Many metrics unavailable for commercial routers (Clarification)**: Only GPT-5 behaves as a true black box, so we exclude it for metrics that require model selections for each input. But for the other routers (e.g., Azure-Router and NotDiamond), we are able to calculate all scores.
    > Reviewer `ZQTb (2): W2/Q5`
3) **Robustness implementation mismatch (Robustness results updated)**: We aligned the evaluation procedure with our robustness definition and updated the result in Figure 9.
    > Reviewer `ZQTb (2): W4/Q2`
4) **Additional routing demand and metrics (Code agent included and new metrics added)**: Coding agent scenarios are already included; other mentioned scenarios haven't been widely supported by today's routers. We also include additional metrics like long-context ability and average generation length. Both provide additional measuring features for users to leverage among routers with similar accuracy and cost.
    > Reviewer `vKx3 (2): W2.2, W4`
5) **Fix cost normalization bias & beta value (Clarification, new results)**: The cost boundary and the beta value are tunable. We provided the leaderboard under different beta values.
    > Reviewer `T3An (6): W1`

---

[Continue on the next page]

---

> ### Author Response · Authors · 2025-11-29
> **[Cont.] Summary for AC**
>
> 6) **Lack of open-ended query, user-experience-related signals (Clarification)**: There is no widely accepted, reliable automatic metric for open-ended queries and user-experience-related signals. We used objective questions to ensure reproducibility and reliability. Also, prior router benchmarks use objective tasks just as we do.
>     > `Reviewer T3An (6): Q2`, Reviewer `vKx3 (2): W1`
> 7) **Heterogeneous model pools (Clarification)**: Model pool is critical for routers, and as a leaderboard, we should encourage routers to explore various models to boost the accuracy-cost tradeoff. We also provided model-pool–invariant metrics to evaluate algorithmic quality (e.g., Optimality scores and normalized deferral plot).
>     > Reviewer `LQ8R (8): W1`, Reviewer `vKx3 (2): W3`
> 8) **Lack rubric-based partial credit, multi-turn utility (Clarification)**: We clarify that rubric-based partial credit is already included in our framework. We view multi-turn utility as our future work; nearly all existing LLM routers are designed and evaluated in a single-turn setting.
>     > Reviewer `T3An (6): Q1`
> 9) **Dataset size limitations (Clarification)**: The current dataset of 8,400 examples is a subset of the original collection of more than 30,000 queries. It offers a great trade-off between domain/difficulty coverage and evaluation speed. The dataset can be easily extended using our proposed principles and the original dataset.
>     > Reviewer `LQ8R (8): W2`
> 10) **Limited contribution (Clarification)**: The reviewer didn't give specific reasons about this. However, based on their and other reviewers' strengths, people agree that our work has significant value. We clarify our contribution in the response.
>     > Reviewer `ZQTb (2): W1`

---

### Meta-Review · Area_Chair_kGhn · 2026-01-05

**Summary:**

RouterArena proposes an open, “live” evaluation platform and leaderboard for LLM routers, motivated by the growing need to select among heterogeneous LLMs under accuracy–cost–latency trade-offs. The submission contributes (i) a principled 8.4k-query dataset spanning domains/categories with stratification, (ii) multi-axis metrics (accuracy, cost, routing optimality, robustness, latency/overhead, etc.), and (iii) an automated pipeline intended to support both open-source and commercial routers and keep a leaderboard updated. Reviewers broadly agree the problem is timely and the systemization is valuable, but raised concerns about cost normalization sensitivity, LLM-as-judge/difficulty labeling validity, robustness definition vs implementation, lack of difficulty-wise analysis, benchmarking philosophy (online vs offline), dataset scale, heterogeneous model pools, and coverage gaps (open-ended/multi-turn/agentic routing and UX signals). The rebuttal addressed most methodological concerns via added analyses (beta sensitivity, human validation of Bloom labeling, empirically-defined difficulty, updated robustness protocol, difficulty breakdowns) and clarified scope/design trade-offs (objective tasks for reproducibility; multi-turn as future work; online leaderboard complementary to offline corpora; pool-invariant “optimality/deferral” metrics).

**Reviewer Concerns:**

Addressed:
Cost normalization / β sensitivity
LLM-as-judge validity
Robustness mismatch
No analysis by difficulty
Commercial-router metric availability
Heterogeneous pools comparability
Additional metrics

Perhaps still missing:
Coverage gap for open-ended, subjective, multi-turn/agentic routing and UX signals (T3An Q2, vKx3): delegated to “future work
Leaderboard “meaningfulness” under heterogeneous pools: pool-invariant metrics may help
Evaluation of true end-to-end latency for commercial systems remains difficult

**Reviewer Scores:**

LQ8R: 8 to 8: was already positive; concerns addressed/clarified
T3An: 6 to 7: β sensitivity + human study + clarification on partial credit/multi-turn likely improves comfort.
ZQTb: 2 to (perhaps) 5?: key complaints, i.e. difficulty analysis, robustness mismatch, commercial metrics, labeling validity, were directly addressed
vKx3: 2 to 4: some concerns resolved but might remain skeptical

This is a difficult one but the score would be overall accept.

---

### Decision · Program_Chairs · 2026-01-26

Accept (Poster)